# Quantitative proteomics defines mechanisms of antiviral defence and cell death during modified vaccinia Ankara infection

Jonas D. Albarnaz [1,2,7,8] ✉, Joanne Kite [1,2,8], Marisa Oliveira[1,2,8], Hanqi Li[1,2], Ying Di[1,2], Maria H. Christensen [3], Joao A. Paulo [4], Robin Antrobus[1,2], Steven P. Gygi [4], Florian I. Schmidt [3], Edward L. Huttlin [4], Geoffrey L. Smith [5,6] & Michael P. Weekes [1,2] ✉

Modified vaccinia Ankara (MVA) virus does not replicate in human cells and is the vaccine deployed to curb the current outbreak of mpox. Here, we conduct a multiplexed proteomic analysis to quantify >9000 cellular and ~80% of viral proteins throughout MVA infection of human fibroblasts and macrophages. >690 human proteins are down-regulated >2-fold by MVA, revealing a substantial remodelling of the host proteome. >25% of these MVA targets are not shared with replication-competent vaccinia. Viral intermediate/late gene expression is necessary for MVA antagonism of innate immunity, and suppression of interferon effectors such as ISG20 potentiates virus gene expression. Proteomic changes specific to infection of macrophages indicate modulation of the inflammatory response, including inflammasome activation. Our approach thus provides a global view of the impact of MVA on the human proteome and identifies mechanisms that may underpin its abortive infection. These discoveries will prove vital to design future generations of vaccines.

Monkeypox virus (MPXV), the cause of the mpox disease, is a zoonotic orthopoxvirus endemic in Central and West Africa[1,2]. Since May 2022, thousands of cases of mpox have been reported in >90 non-endemic countries worldwide, with sustained human-to-human transmission[3]. MPXV causes a smallpox-like illness, with severe disease seen in immunocompromised individuals, children and pregnant women. The ongoing outbreak has been caused by MPXV clade 2b (within the formerly designated "West African" clade)[4]. Although mortality is commoner with MPXV clade 1 (formerly designated "Central African"

or "Congo Basin" clade), 0.1% of infections with clade 2 MPXV have resulted in death, with increased risk in the context of HIV co-infection[5].

Although the World Health Organisation (WHO) has not advised mass vaccination, post-exposure prophylaxis is recommended for higher-risk mpox contacts, and pre-exposure prophylaxis for at-risk healthcare workers[3]. Highly effective protection can be provided by live vaccines originally developed against smallpox, which was itself eradicated through vaccination with a related orthopoxvirus,

[1]Cambridge Institute for Medical Research, University of Cambridge, Cambridge CB2 0XY, UK. [2]Department of Medicine, University of Cambridge, Cambridge CB2 0XY, UK. [3]Institute of Innate Immunity, University of Bonn, 53127 Bonn, Germany. [4]Department of Cell Biology, Harvard Medical School, 240 Longwood Avenue, Boston, MA 02115, USA. [5]Department of Pathology, University of Cambridge, Tennis Court Road, Cambridge CB2 1QP, UK. [6]Sir William Dunn School of Pathology, University of Oxford, South Parks Road, Oxford OX1 3RE, UK. [7]Present address: The Pirbright Institute, Ash Road, Pirbright, Woking GU24 0NF, UK. [8]These authors contributed equally: Jonas D. Albarnaz, Joanne Kite, Marisa Oliveira. ✉e-mail: jonas.albarnaz@pirbright.ac.uk; mpw1001@cam.ac.uk

vaccinia virus (VACV)[6]. The most recent generation of smallpox vaccines derive from modified vaccinia Ankara (MVA), itself derived by >570 passages of the VACV strain chorioallantois vaccinia Ankara (CVA) in chicken embryo fibroblasts[7–9]. Serial passage resulted in loss of ~30 kb of the genome, and loss of replicative capacity in human cells[10]. Because MVA has been shown to be safe and immunogenic in both healthy and immunocompromised individuals[11,12], it has also been investigated extensively as a vaccine vector for viruses including Ebola, respiratory syncytial virus, HIV, and SARS-CoV-2[13]. Vaccination with MVA has also been shown to induce MPXV neutralising antibodies in humans[14] and is protective in a non-human primate model[15].

The biological effects of MVA at the cellular level are poorly understood with no prior systematic proteomic investigation. Previous transcriptomic analyses of MVA and VACV-WR infection of the transformed HeLa cell line indicated that MVA induced a number of changes in the host transcriptome, including changes in the immune regulation and NF-κB signalling[16]. More recently, a single-cell transcriptomic (scRNAseq) analysis of dendritic cells infected with MVA revealed two possible outcomes: sensing of infection via cGAS-STING leading to production of inflammatory cytokines and activation of uninfected bystanders, or induction of caspase activity and cell death via apoptosis[17]. Previously, we conducted a systematic quantitative temporal proteomic analysis of VACV-WR-infected human fibroblasts to elucidate the functional consequences of VACV-induced dynamic changes in the host proteome[18]. However, little information is available about the consequence to the host proteome of non-productive, abortive infections.

In this study, we therefore conduct a comprehensive proteomic analysis of MVA and host throughout the whole course of infection, including inactivated controls to understand the contribution of the viral particle with no-, or limited viral gene expression. We show extensive remodelling of the human proteome in both fibroblasts and macrophages following MVA infection. Despite attenuation, MVA exerts antagonism of the innate immune response, which might contribute to its potent capacity to express high levels of heterologous antigens. We also show that MVA activates the inflammasome and induces pyroptosis in human macrophages, despite retaining proteins that inhibit both processes.

## Results

### Quantitative temporal analysis of MVA infection

To build a global picture of changes in host and viral proteins throughout the course of MVA infection, we infected telomerase reverse transcriptase (TERT)-immortalised primary human foetal foreskin fibroblasts (HFFF-TERTs) with MVA at a multiplicity of infection (MOI) of 5 in biological duplicate. HFFF-TERTs have been well established as a model for a variety of different viral infections[18–21], and MOI 5 infected >96% of cells (Supplementary Fig. 1a). Application of 16-plex TMT and triple-stage mass spectrometry (MS3) with real-time search (RTS)[22,23] quantified protein expression over the full course of infection (Fig. 1a). In the same experiment, we included identical analyses with heat-inactivated MVA (MVA-HI), or the viral DNA replication inhibitor cytosine arabinoside (AraC), to understand the contribution of the viral particle with no gene expression, or expression of early viral genes only, respectively (Fig. 1a).

7545 human and 144 viral proteins were quantified, providing a global view of changes in protein expression during infection. The greatest magnitude of protein fold change occurred late during infection, and heat inactivation ablated the majority of host protein downregulation (Fig. 1b, Supplementary Fig. 2). All data are shown in Supplementary Data 1, including the worksheet "Plotter". This tool generates expression temporal profiles for each of the human or viral proteins quantified.

## MVA regulates multiple mediators of intrinsic, innate and adaptive immunity

Over 18 h of infection, 383 human proteins were downregulated >2-fold, and 66 human proteins upregulated >2-fold. Interestingly, Database for Annotation, Visualisation and Integrated Discovery (DAVID) software[24] revealed that diverse groups of cell surface proteins were downregulated early during MVA infection including multiple NK and T-cell ligands suggesting that a key focus of the first phase of infection may be evasion of cellular immunity (Fig. 2a, b). Upregulated proteins included inflammatory mediators such as cyclooxygenase 2 (PTGS2) and lactotransferrin (LTF)[25] (Supplementary Data 2). To distinguish with high confidence host factors that were downregulated by viral gene expression at distinct stages of infection, we applied a series of filters that identified differential modulation of host protein expression in the presence of unmodified MVA, MVA+AraC and MVA-HI (Fig. 2c, Supplementary Fig. 3b–d, Supplementary Data 2). Of particular interest, viral proteins expressed later during infection exquisitely regulated multiple components of the nuclear pore complex (NPC). This was interesting since nuclear pore proteins may be essential for replication of poxviruses[26] and a recent study showed the antiviral restriction factor FAM111A cleaves NUP62 during MVA infection[27]. Proteins downregulated by early-expressed genes included secreted inflammatory mediators such as metallopeptidase inhibitor 2 (TIMP2) and transforming growth factor β−1 (TGFB1) (Supplementary Fig. 3c), and the antiviral restriction factor TRIM5 (Supplementary Data 1). TRIM5 restricts orthopoxviruses and is targeted for proteasomal degradation by the viral protein C6, which is expressed by MVA[28]. Infection with heat-inactivated virus offered the opportunity to identify proteins that were upregulated upon sensing of the viral particle, but whose expression was limited by viral genes (Fig. 2c, lower panels). These included four poly ADP-ribose polymerase (PARP) proteins including Zinc Finger Antiviral Protein (ZC3HAV1, ZAP), a variety of proteins with key roles in immunity including the viral DNA sensor IFI16, interferon regulator TRIM26 and interferon-stimulated genes including IFIT2, ISG20 and OASL (Fig. 2c, lower panels). Certain factors were downregulated in all three conditions, including extracellular matrix components and multiple collagens (Supplementary Fig. 3b), suggesting that these changes may represent a cellular response to infection, as opposed to being induced by viral gene expression.

To understand better how proteins altered during various stages of MVA infection relate to one another and to the larger proteome, we superimposed groups of proteins displaying each characteristic abundance signature (Fig. 2d) onto the BioPlex network of human protein-protein interactions[29]. Graph assortativity was then calculated, measuring the tendency of proteins in a particular group to interact preferentially with each other compared to proteins that are not part of the selected group (Supplementary Fig. 3d). Proteins downregulated during later stages of MVA infection showed a strong tendency to self-associate, including nuclear pore components, and a selection of nucleolar proteins associated with ribosome biogenesis (e.g. SURF6, RBM28, RPL7L1, ZC3HAV1, CDK105) (Fig. 2d).

To gauge the functional significance of the host proteomic changes during MVA infection, we performed a small-scale gain-of-function screen via ectopic expression in HFFF-TERTs (Supplementary Fig. 4a, b). We included a selection of proteins that were downregulated during MVA infection (GLE1, NSA2, NUP54, NUP62, NUP88, RBM28). We also examined IFN-stimulated proteins whose induction was suppressed by infection, but not by heat-inactivated virions or upon inhibition of viral DNA replication (ISG20, OASL, ZNFX1). Cells were infected with an MVA strain expressing GFP under the control of a viral early/late promoter (MVA-GFP)[30] as a proxy of viral gene expression. Out of the nine selected proteins, only ISG20 significantly altered MVA-GFP infection (Fig. 2e). Of note,

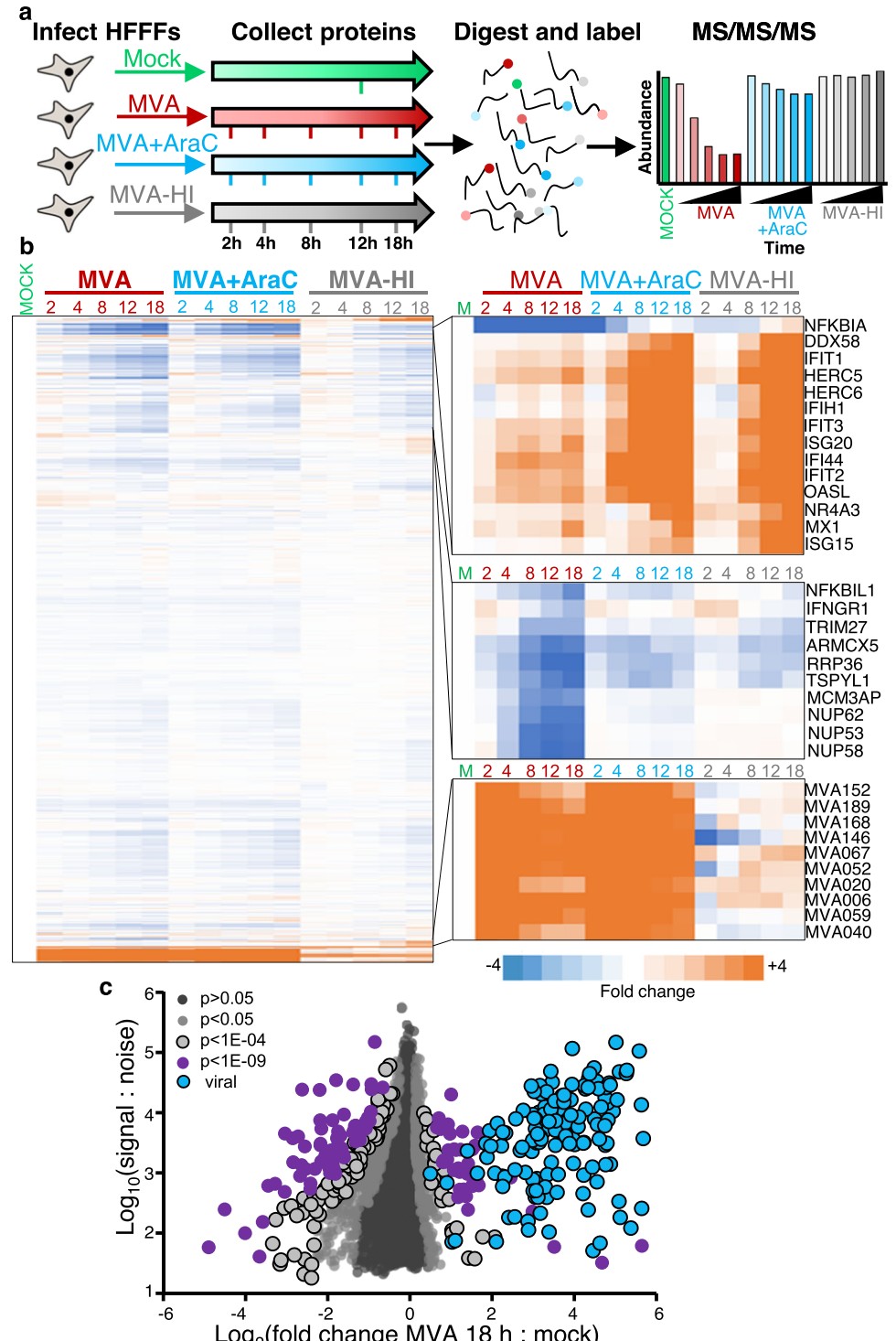

**Fig. 1 | Quantitative temporal proteomic analysis of MVA infection. a** Schematic indicating the experimental workflow. **b** Hierarchical clustering of all proteins quantified in the two biological repeats. An enlargement is shown indicating groups of proteins that were significantly down- or upregulated during the course of the experiment. **c** Scatterplot of all proteins quantified at 18 h of infection, showing average fold change. *P*-values were estimated using Significance B values, and corrected for multiple hypothesis testing using the Benjamini–Hochberg method. Significance A and B tests are two-sided tests that allow for non-equal left and right-sided standard deviations[95, 96]. Dark grey: *p* > 0.05, Light grey: *p* < 0.05, Light grey with border: *p* < 1E−04, Purple: *p* < 1E−09, Blue: viral. Source data are provided as a Source Data file.

it is possible that other proteins expressed in this screen at a level lower than ISG20 including GLE1, NSA2 and ZNFX1 might nevertheless exhibit an effect (Supplementary Fig. 4a, b). Although ectopic expression of ISG20 limited virus-driven GFP expression, loss-of-function studies will need to be conducted to confirm this phenotype.

## Differential regulation of the host proteome by MVA and virulent VACV

To discover what may underpin the differences in pathogenesis and replication between MVA vaccine, which is avirulent and replication-incompetent in human cells, and the mouse-adapted, virulent, and replication-competent VACV strain Western Reserve (VACV-WR), we

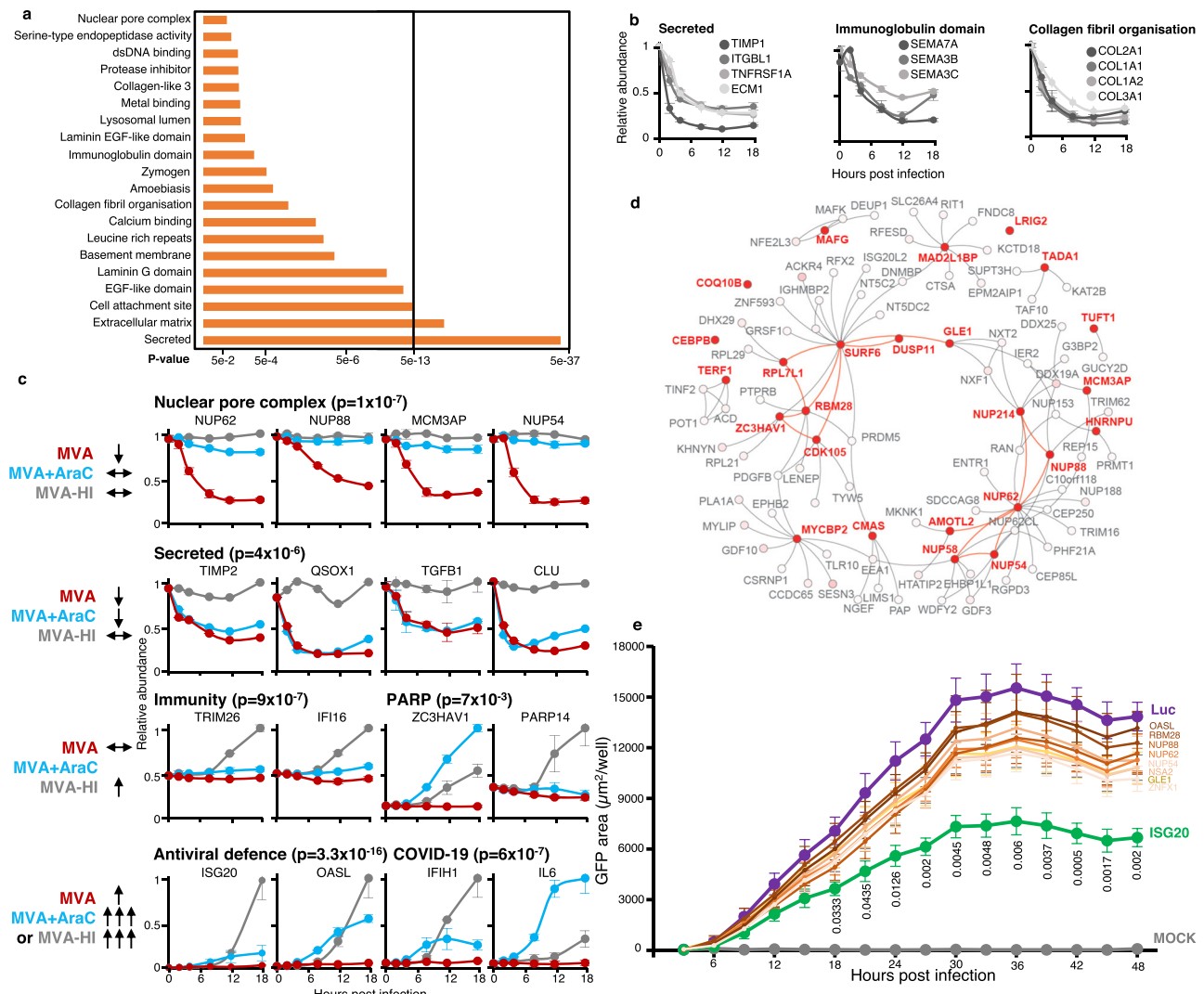

**Fig. 2 | MVA regulates multiple mediators of intrinsic, innate and adaptive immunity. a** Functional enrichment analysis of proteins downregulated >2-fold at ≥1 time point during MVA infection of HFFF-TERT cells. *P*-values were estimated in DAVID software using Fisher's Exact tests and were corrected for multiple hypothesis testing using the Benjamini−Hochberg method. **b** Example components from cell surface-related pathways identified in **a**. *n* = 2 biological replicates from independent experiments. Data are presented as mean values ± range. **C** Functional enrichment analysis of groups of proteins regulated as indicated in the left-hand column, and examples of components of enriched pathways. Full data is shown in Supplementary data 2. 2-fold is used throughout the manuscript as a cutoff for downregulation, with proteins unchanged defined as downregulated <1.25 fold at all time points measured. *n* = 2 biological replicates from independent experiments. Data are presented as mean values ± range. *P*-values were estimated in DAVID software using Fisher's Exact tests and were corrected for multiple

hypothesis testing using the Benjamini−Hochberg method. **d** Plot of interactions among proteins decreased upon infection with MVA but unchanged in MVA+AraC and MVA-HI samples (red text = proteins connected by red edges). For context, an additional 74 neighbouring proteins are shown (grey text and dots). Relative proximity to the proteins that were decreased upon infection with MVA but unchanged in MVA+AraC and MVA-HI samples is indicated by red shading, and was quantified via random walk with restart (Methods). **E** HFFF-TERT cell lines stably expressing the proteins indicated were infected with MVA-GFP at MOI 5. At the indicated times, GFP expression was measured and compared to infected Luciferase (Luc)-expressing cells. Significance calculated using two-way ANOVA followed by post-hoc Dunnet's multiple comparisons test. Means ± SEM (*n* = 8 biological replicates per condition, from 4 independent experiments) and *p*-values < 0.05 are shown. Source data are provided as a Source Data file.

next characterised similarities and differences in host regulation. We compared results from the present study with our prior proteomic analysis of VACV-WR[18], which is the reference virus strain for studies with orthopoxviruses, is highly neurotropic in mice, and was derived from the New York City Board of Health first-generation smallpox vaccine. Whereas at least 30% of proteins were co-regulated by both viruses, there were a number of key differences (Fig. 3a, b, Supplementary Fig. 5a, b, Supplementary Data 3). Proteins substantially downregulated during MVA infection but minimally modulated by VACV-WR were enriched in interacting components of the nuclear pore complex and certain glycoproteins. The latter included multiple immune modulators such as semaphorins, which modulate

intercellular communication between immune cells[31], transforming growth factors TGFB1 and TGFB2, and the endoplasmic reticulum stress sensor EIF2AK3 (also known as PERK) (Fig. 3a−c, Supplementary Fig. 5b, d). MVA, but not VACV-WR, also induced the downregulation of negative (NFKBIA and NFKBIB, also known as IκBα and IκBβ, respectively) and positive (BIRC2, also known as cIAP1) regulators of NF-κB immune signalling (Fig. 3a, Supplementary Fig. 5b, d). IκBs sequester inactive NF-κB in the cytoplasm and their proteasomal degradation releases NF-κB components to accumulate in the nucleus where they activate the expression of immunity-related genes including interferons[32]. Therefore, the down-regulation of NFKBIA and NFKBIB indicate activation of NF-κB during MVA, but not VACV-WR, infection

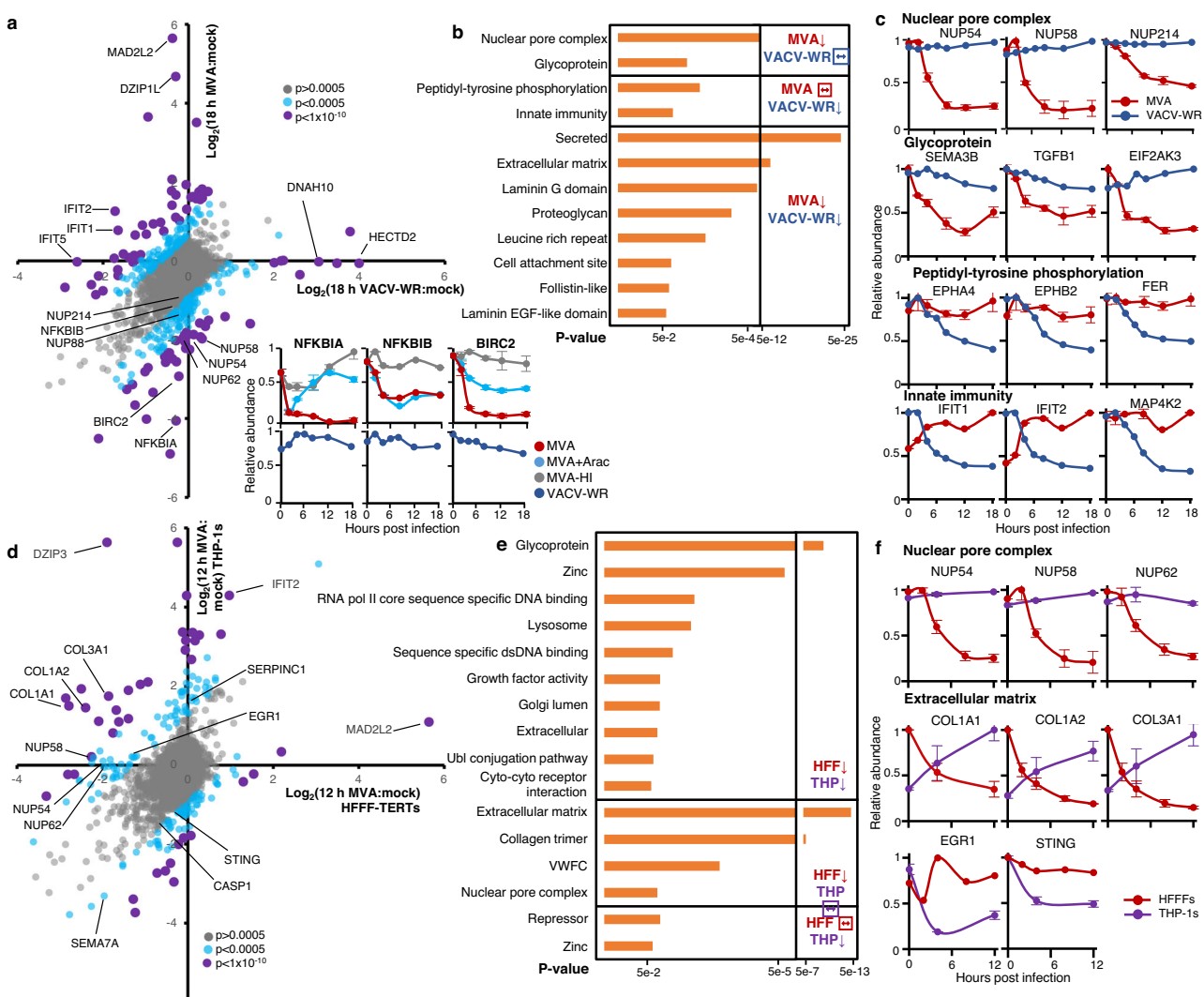

**Fig. 3 | Differential protein changes during infection of human fibroblasts with MVA or VACV-WR, and during MVA infection of HFFFs or THP−1 cells.**
**a** Scatterplot of all proteins quantified at 18 h of infection with both MVA and VACV-WR. *P*-values were estimated using Significance A values (described in the legend to Fig. 1c), and corrected for multiple hypothesis testing using the Benjamini–Hochberg method[95,96] and are indicated on the plot via colouring of the dots as indicated in the legend. Example profiles for NF-κB pathway proteins are shown in the lower panels. *n* = 2 biological replicates from independent experiments. Data are presented as mean values ± range. Grey: *p* > 0.0005, Blue: *p* < 0.0005, Purple: *p* < 1E−10.
**b** Functional enrichment analysis of proteins (i) downregulated by MVA but not VACV Western Reserve (VACV-WR), (ii) downregulated by VACV-WR but not MVA, or (iii) downregulated by both MVA and VACV-WR. *P*-values were estimated employing the Fisher's Exact tests and were corrected using the method of Benjamini–Hochberg. **c** Example components from pathways identified in **b**. *n* = 2 biological replicates from independent experiments. Data are presented as mean values ± range. **d** Scatterplot of all proteins quantified at 12 h of infection in HFFF-TERTs or THP-1s. *P*-values were estimated and displayed as in **a**. For ease of visualisation, fold upregulation was limited to a maximum of 50. Grey: *p* > 0.0005, Blue: *p* < 0.0005, Purple: *p* < 1E−10. **e** Functional enrichment analysis of proteins regulated as indicated. *P*-values were estimated as in **a**. **f** Example components from pathways identified in **e**. *n* = 2 biological replicates from independent experiments. Data are presented as mean values ± range. Source data are provided as a Source Data file.

as previously described[33] and highlights that our proteomic analysis captures known aspects of virus sensing. The presence of NFKBIA re-synthesis during infection with MVA+AraC but not MVA infection also indicates a MVA blockade of the NF-κB pathway at or downstream of IκBs degradation. Alternatively, NFKBIA re-synthesis could indicate that a protein, or proteins, expressed late in MVA infection results in the degradation of IκB, which is abrogated in the presence of AraC. Conversely, proteins down-regulated during VACV-WR infection but minimally modulated by MVA were enriched in factors involved in tyrosine phosphorylation and innate immunity. The former pathway included members of the Ephrin family of receptor tyrosine kinases (EPHA4, EPHB2, and EPHB3) and the proto-oncogene FER whereas the latter included multiple interferon-stimulated genes and MAP4K2, an upstream activator of mitogen-activated protein kinases, key

regulators of immune signalling downstream of pathogen sensing[34] (Fig. 3b, c, Supplementary Fig. 5b).

## MVA regulates multiple cell death-related proteins and activates inflammasomes in macrophages

Multiple cell types are infected with MVA at the site of inoculation, including professional antigen-presenting cells that express high levels of viral antigens[35]. Therefore, to analyse MVA-induced changes in the proteome of antigen-presenting cells and to test the generality of our findings in human fibroblasts, we repeated the proteomic analysis in phorbol 12-myristate 13-acetate (PMA)-differentiated THP-1 macrophages (referred henceforth to as THP-1s; Supplementary Fig. 6). 8104 human proteins in total were quantified in THP-1s, of which 19% were not quantified in HFFF-TERTs. Proteins downregulated >2-fold in

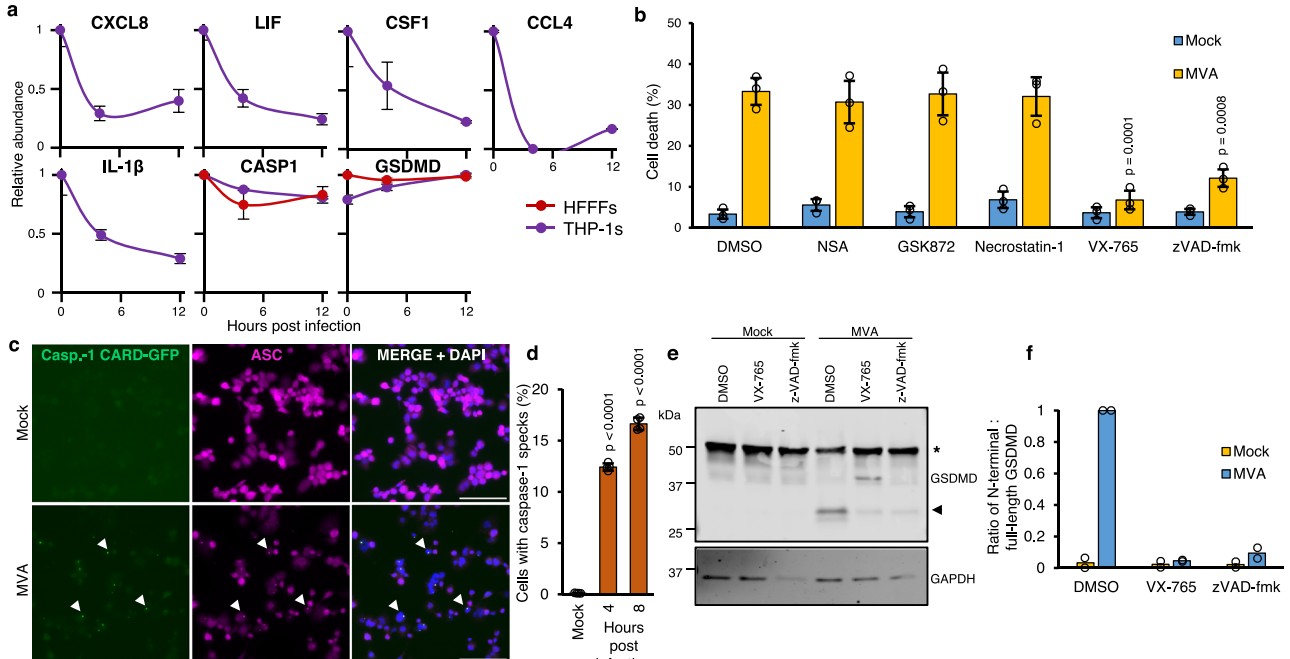

**Fig. 4 | MVA infection induces pyroptosis in THP-1 cells. a** Representative temporal profiles of secreted proteins and proteins involved in the inflammasome pathway during MVA infection. (continued Supplementary Figure. 7b). $n = 2$ biological replicates from independent experiments. Data are presented as mean values ± range. **b** Cell death assessed by lactate dehydrogenase release assay. PMA-differentiated THP-1 macrophages were treated with the indicated inhibitors and infected with MVA (MOI = 5) for 8 h. Means ± s.d. ($n = 3$ independent experiments) are shown. **c** Immunofluorescence to visualise inflammasome assembly (caspase-1/ASC speck formation). Differentiated THP-1 cells that ectopically express CARD1-GFP (THP-1C1C-GFP) were infected with MVA (MOI = 5) for 8 h in presence of caspase-1 inhibitor VX-765 (4-h infection shown in Supplementary Fig. 8). Representative micrographs show caspase-1 CARD-GFP (green), ASC immunostaining (magenta) and DAPI staining (blue). Scale bar = 100 μm. Arrowheads indicate caspase-1/ASC specks. **d** Quantification of cells with caspase-1 CARD-GFP specks from **c**. Means ± s.d. ($n = 4$ biological replicates from 2 independent experiments) are shown. **e** Immunoblot showing gasdermin D (GSDMD) cleavage is stimulated during MVA infection. Differentiated THP-1s were treated as indicated and infected with MVA (MOI = 5) for 8 h. **f** ratio of cleaved N-terminal fragment (arrowhead) over full-length GSDMD (indicated by an asterisk) band intensities from **e**. The positions of molecular mass markers in kDa are shown on the left of immunoblots. Means ($n = 2$ independent experiments) are shown. Significance in **b** and **d** was calculated using one-way ANOVA followed by post-hoc Dunnett's multiple comparisons test. Source data is provided as a Source Data file.

infected THP-1s were enriched in cytokines, chemokines, and other secreted factors with immune regulatory functions (Supplementary Fig. 7, Supplementary Data 2). These included the macrophage colony-stimulating factor 1 (CSF1), leukaemia inhibitory factor (LIF), interleukin 1β (IL-1β), and the chemokines C-X-C motif chemokine ligand 10 (CXCL10), CXCL8, and C-C motif chemokine 4 (CCL4). In contrast, heat-inactivated MVA stimulated the accumulation of these secreted factors (Supplementary Figs. 6b and 7c). Possible explanations for this downregulation at the level of the whole cell lysate include specific protein degradation or transcriptional downregulation, or infection-stimulated production and subsequent release of inflammatory cytokines and chemokines[17,36–38]. The reduced abundance of cell-associated IL-1β during MVA infection of THP-1s is a hallmark of inflammasome activation, proteolytic processing of pro-IL-1β and pyroptotic cell death (Supplementary Fig. 7c)[39].

Inflammasomes are multimeric protein complexes that assemble in response to infectious or cellular damage stimuli and activate pro-inflammatory caspases, most notably caspase-1. Active caspase-1 cleaves pro-IL-1β, pro-IL-18, and gasdermin D (GSDMD)[39]. To determine whether inflammasomes are indeed activated during MVA infection of THP-1s, we first measured cell death by lactate dehydrogenase (LDH) release assay. Infection of THP-1s with MVA induced a sharp increase in cell death, which was reversed by treatment with a caspase-1 inhibitor (VX-765) and, to a lesser extent, the pan-caspase inhibitor (zVAD-fmk) (Fig. 4a). Conversely, inhibitors of the necroptosis mediators RIPK1, RIPK3, and MLKL did not affect MVA-induced cell death at concentrations that blocked necroptosis in HFFF-TERTs (Fig. 4a, Supplementary Fig. 8a). Next, we

sought to visualise and quantify inflammasome assembly in infected THP-1 macrophages engineered to express caspase-1 caspase recruitment domain (CARD) fused to EGFP (C1C-EGFP)[40]. Following infection with MVA in the presence of VX-765 to avoid cell loss due to pyroptosis, C1C-GFP redistributed to single specks in each cell at 4 and 8 h p.i. (Fig. 4b, c; Supplementary Fig. 8a). Co-localisation with the adaptor protein apoptosis-associated speck-like protein containing a CARD (ASC) confirmed the identity of the specks as inflammasomes (Fig. 4b; Supplementary Fig. 8a). ASC specks were also observed following infection of parental THP-1s with MVA (Supplementary Fig. 8b, c). Lastly, we also observed cleavage of GSDMD downstream of inflammasome activation by MVA infection. VX-765, as well as zVAD-fmk, prevented GSDMD cleavage, indicating its caspase-1-dependence (Fig. 4d, e).Multiple mechanisms of cell death have been described in MVA-infected macrophages, such as apoptosis and necroptosis in a manner largely dependent on STING-mediated sensing of viral DNA[36]. We observed differential modulation of apoptosis-related genes in THP-1 macrophages infected with MVA, with upregulation of pro-apoptotic NOXA (PMAIP1) and downregulation of anti-apoptotic MCL1 (Supplementary Fig. 7c). The abundance of other important mediators (BAX, BAK1) and regulators (BAD, BCL2L1) of the mitochondrial pathway of apoptosis remained unchanged. As in HFFF-TERTs, infection of THP-1 macrophages also promoted downregulation of cIAP1 (BIRC2), a key regulator of the TNF-dependent extrinsic pathway of apoptosis (Supplementary Fig. 7c). Furthermore, it was striking to observe that STING (TMEM173) was downregulated during MVA infection of THP-1, but not HFFF-TERT cells (Fig. 3f), which is consistent with STING

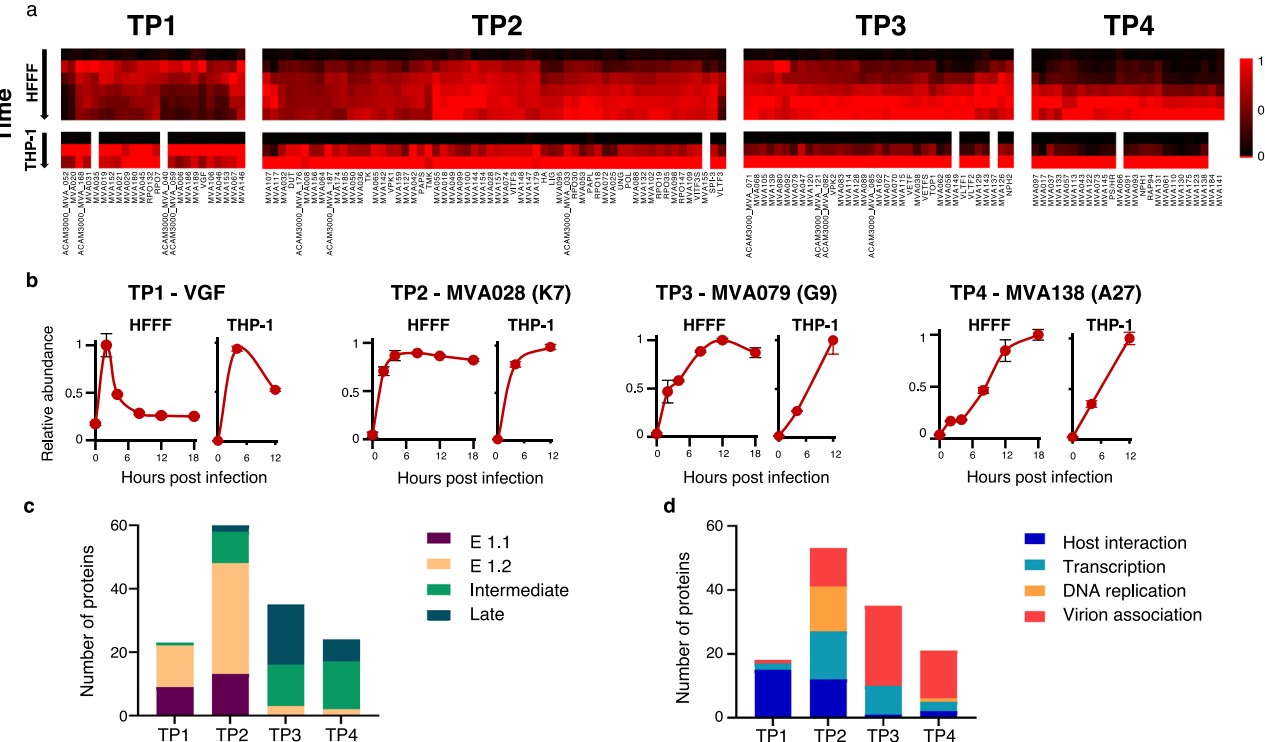

**Fig. 5 | Temporal classes of MVA protein expression. a** Viral proteins were clustered into 4 classes defined by the k-means method. Proteins were subsequently clustered hierarchically within each class. **b** Examples of each class. Protein names in brackets indicate orthologous proteins from VACV-WR (Supplemental Data 4). *n* = 2 biological replicates from independent experiments. Data are presented as mean values ± range. **c** Comparison of MVA viral protein and VACV transcript classes in HFFF. **d** Functional classification of MVA viral proteins, based on information from[47] for VACV in HeLa cells. Source data is provided as a Source Data file.

degradation following activation by MVA, to prevent sustained innate immune signalling[41].

## Comparison of protein changes during MVA infection in fibroblasts and macrophages

In general, there was good correspondence between protein changes in HFFF-TERTs and THP-1s, albeit with subtle differences in the magnitude of change in each cell type. 70% of proteins downregulated >2-fold in either HFFF-TERTs or THP-1s were also downregulated at least 1.25-fold in the other cell type, whereas 18% were only downregulated in HFFFs and 12% only in THP-1s (Fig. 3d, e, Supplementary Data 1). Multiple pathways were also commonly regulated (Fig. 3e, Supplementary Data 3). To aid searching of this data, a comparison of protein up/downregulation in each cell type is included in Supplementary Data 7.

Differences in regulation between cell types included the downregulation of members of the nuclear pore complex in HFFF-TERTs, but not in THP-1s, and the downregulation only in THP-1s of the transcriptional regulator early growth response 1 (EGR1), which is a host-dependency factor for VACV downstream of mitogen-activated protein kinases[42,43] (Fig. 3e, f). Also, whereas collagen proteins were downregulated during infection of HFFF-TERTs, the same proteins were upregulated in THP-1s (COL1A1, COL1A2, COL3A1, COL5A1, and COL12A1, Fig. 3f). MVA-infected THP-1s also exhibited a global increase in the abundance of factors of the complement (C4A, C5, C7, CFH, and CFI) and coagulation (F2, F5, F10, antithrombin III (SERPINC1), and β-fibrinogen (FGB)) pathways, unlike in HFFF-TERTs, in which they were in general minimally regulated (Supplementary Fig. 7c). Blockade of DNA replication did not affect the upregulation of collagens and complement/coagulation proteins, whereas heat-inactivated MVA was unable to induce these factors to the same extent as infectious MVA.

## Temporal analysis of MVA protein expression to inform viral-host interactions

High-definition temporal analysis of viral protein expression can facilitate direct correlation between viral and cellular protein profiles to give insights into virus-host protein interaction[18]. We quantified 77% and 74% of all predicted MVA proteins in HFFF-TERTs and THP-1s, respectively. To define different classes of protein expression, we employed k-means clustering and included data from HFFF-TERT samples treated with AraC. This suggested that there are at least four distinct temporal protein profiles of viral protein expression in both cell lines tested (Fig. 5b, Supplementary Fig. 9a).

Some of the earliest Tp1 (temporal profile 1) and Tp2 classes of viral protein exhibited increased expression in the presence of AraC (Supplementary Fig. 9b, c), likely explained by the absence of negative regulation from post-replicative proteins. In contrast, the Tp3-class proteins were partially inhibited by AraC, with expression of Tp4 class proteins almost completely inhibited (Supplementary Fig. 9b, c). Tp4 class proteins included multiple VACV envelope proteins, including B5 and A27 homologues PS/HR and MVA138 that encode targets of virus-specific T-cell memory[44] (Supplementary Data 4). Tp1-class proteins included MVA102 (VACV-WR D5), which uncoats viral genomes early during infection[45], and MVA019 (VACV-WR C6) (Fig. 5a). Previously, we demonstrated that C6 targets host restriction factors HDAC4, HDAC5, and TRIM5 for proteasomal degradation[18,28,46]. This classification thus serves as a particularly useful resource to predict interactions between host factors downregulated at early, intermediate or late times after infection and their viral counterparts and may help to form a rational basis for prediction of the effects of mutations introduced in individual genes during vaccine production by additional passage, plaque purification or insertion of foreign genes derived from other pathogens.

To correlate protein function with temporal class, we annotated the quantified MVA proteins with functions and transcriptional

temporal classes previously employed in[47,48]. The comparison between transcript and protein data was conspicuous: 22/23 Tp1 proteins were encoded by E1.1 or E1.2 class transcripts, and 22/24 Tp4 proteins by I or L transcripts ($p < 0.0001$, Fisher's exact test). The Tp2 proteins were mostly encoded by E1.2 transcripts (Fig. 5c). Such overlap between different transcriptomic and proteomic studies reinforces the biological relevance of the protein-based temporal classification of MVA gene expression and emphasises that temporal control of orthopoxvirus gene expression is predominantly at the transcriptional level (Supplementary Data 4). Proteins annotated with the category "host interaction" were mainly expressed early during infection (Tp1-2); "DNA replication" functions were mainly Tp2-class, and structural proteins or proteins involved in virion assembly primarily related to the Tp3 and Tp4 classes (Fig. 5d).

## Discussion

Upon infection of human cells, MVA does not produce infectious new virions and is therefore classed as a replication-incompetent vaccine[9]. In the spectrum of vaccines available for use against human diseases, replication-incompetent vaccines lie between attenuated vaccines and inactivated vaccines, without the safety concerns of the former or the reduced immunogenicity of the latter[49]. Despite an ultimately abortive MVA infection of human cells, the viral life-cycle is typical of other poxviruses until early morphogenesis, including early gene expression, genome uncoating, DNA replication, and post-replicative gene expression[50]. Our global analysis of MVA infection highlighted a pervasive modulation of human host proteins and processes. Such an understanding, particularly including regulation of functions related to immunity, is critical for a complete assessment of MVA's applicability and safety as a vaccine against MPXV and other orthopoxviruses, and as a vaccine vector against other pathogens.

It is noteworthy that we used an MVA strain that is different from the MVA-BN strain currently used as a mpox/smallpox vaccine. Both the MVA-BN and our MVA stocks derived from the same MVA stock at passage 575 (MV-575), but MVA-BN was further passaged in chicken embryo fibroblasts[51]. These additional passages are claimed to have further attenuated MVA-BN in comparison with MVA-575, although both strains exhibit very similar biological properties[52,53]. Although it is possible that some of our findings might not apply to MVA-BN, our study of MVA-575 will additionally be of use to future vaccines derived from this and other similarly passaged precursors.

Our proteomic analysis of MVA infection in human fibroblasts and macrophages showed ~2,500 proteins were only quantified in one cell type, underscoring the importance of analysing multiple cell types in order to obtain a high-definition proteomic landscape of human cellular infection. Interferons and other cytokines induced upon viral sensing are key regulatory molecules in innate immunity and inflammation and play important roles in fine-tuning antigen-specific adaptive immunity during viral infection[54,55]. MVA-induced regulation of the interferon response and NF-κB signalling, indicated both by limited upregulation of interferon-stimulated gene (ISG) products and direct regulation of NF-κB, will therefore be likely to impact the immunological memory elicited by vaccination. Interferon regulation is likely to occur at all stages of infection, indicated by the increased response seen upon blockage of post-replicative viral gene expression and in cells exposed to heat-inactivated virions in comparison to cells infected with unmodified MVA. This is in contrast with the antagonism of the interferon response by VACV-WR. For instance, VACV early protein C9 targets ISG products IFIT1, IFIT2, and IFIT3 for proteosomal degradation[56]. Through serial passage in chicken embryo fibroblasts, the MVA genome has accumulated six major deletions and many small insertion and deletions (indels), which, in combination, result in the loss of 25 open reading frames (ORFs) when compared to wildtype VACV strains[10,57]. In addition, 12 ORFs have been split due to mutations, including the ORF encoding a C9 orthologue, explaining the absence

of IFIT degradation during MVA infection[57]. Analysis both of viral protein temporal profiles and inactivating mutations are thus likely to yield candidates antagonists for host factors downregulated during VACV-WR, but not MVA infection including Ephrins and MAP4K2.

Suppression of the IFN response might represent one of the mechanisms responsible for the capacity of MVA vectors to attain high levels of heterologous gene expression under the control of viral promoters[50]. Of three ISGs included in our gain-of-function screen, we could confidently conclude that ectopic ISG20 can limit MVA gene expression whereas ectopic OASL cannot. A 3′–5′ exonuclease with strong preference for RNA, ISG20 can also degrade deaminated viral DNA (reviewed in ref. 58). Genomic analyses support that APOBEC3-mediated deamination of the viral genome is a key mutational driver in the MPXV strain that emerged globally in 2022[4]. A plausible hypothesis is that the degradation APOBEC3-deaminated viral DNA by ISG20 might constitute another host defence against poxviruses. In light of the recent proposal that subset of "dominant" ISGs may confer most of the inhibition of a given virus by IFNs[59], it is not surprising that, out of three ISGs, only ISG20 exerted significant antiviral activity against MVA.

Host proteins with concordant modulation by MVA and VACV-WR might indicate that viral mechanisms of host modulation are still functional in MVA. Examples include common downregulation of the antiviral restriction factors HDAC5 and TRIM5 by VACV-WR and MVA, mediated by VACV protein C6 (MVA019)[18,28]. Proteasomal degradation represents the main mechanism responsible for downregulation of host protein during VACV-WR infection[18] and therefore, most host proteins downregulated during MVA infection are likely also to be degraded by the proteasome. Alternatively, common modulation of protein expression might reflect common cellular responses to infection, such as the downregulation of multiple collagens in fibroblasts.

A total of 101 proteins were downregulated during MVA, but not VACV-WR infection. These included multiple components of the nuclear pore complex, whose downregulation is likely to disrupt nucleocytoplasmic transport[60]. The innate immune response to virus infection requires an intact NPC to mediate nucleocytoplasmic transport of transcription factors and mRNAs and therefore, downregulation/degradation of NPC proteins may be a viral immune evasion strategy. Although some viral proteases inactivate nucleoporins (NUPs) by cleavage[61], NUP abundance did not vary substantially during infection by multiple human pathogenic viruses, such as human cytomegalovirus[19], herpes simplex virus type 1[20], influenza A virus[62], Epstein-Barr virus[63], SARS-CoV-2[64] and HIV[65]. Three of the NUPs downregulated by MVA (NUP54, NUP62, NUP88) were identified as host factors necessary for VACV-WR morphogenesis[26], raising the possibility that NUP downregulation contributes to arrested virion maturation during MVA infection of human cells. It remains to be determined whether this MVA-induced modulation is shared with the parental strain (CVA) or was acquired during the serial passage in chicken embryo fibroblasts. The molecular mechanisms underpinning restriction of MVA in human cells are not fully understood but are largely attributed to the loss or disruption of genes encoding host-range factors[66,67]. Nucleoporins have been implicated in antiviral defences, including as co-factors in the MX2-mediated restriction of HIV-1[68,69] or as substrates for proteolytic cleavage by the antiviral restriction factor FAM111A, which restricts SV40 by disrupting the nuclear pore complex[70]. In line with our observations, a recent report confirmed that FAM111A cleaves NUP62 and disrupts nuclear barrier during MVA infection, in addition to inducing the degradation of the viral single-stranded DNA-binding (SSB) protein I3[27]. The antiviral functions of FAM111A are antagonised by the poxvirus host-range factor SPI-1, which is missing in MVA[27,71]. Restoration of SPI-1 in MVA rescues virus replication in human cells[66]. In our gain-of-function screen, ectopic expression of NUP54, NUP62 or NUP88 did not significantly affect MVA gene expression, possibly because the rescue of individual

nucleoporins is insufficient to restore the functional consequence of the simultaneous downregulation of several nuclear pore complex components.

Studies of immune sensing of MVA infection have focused on antigen-presenting cells. These include dendritic cells and macrophages, which express high levels of MVA proteins at the site of inoculation and present MVA-derived antigens in lymph nodes, underpinning the immunogenicity of MVA-based vaccines[35]. We therefore compared our findings in THP-1 cells with the recent single-cell transcriptomic analysis of human dendritic cells infected with MVA[17]. Döring et al. [17]. made key observations that were corroborated by our proteomic analysis. First, MVA infection triggers type I interferons and inflammatory cytokines downstream of virus DNA sensing by the cGAS-STING-TBK1-IRF3 pathway. Second, bystander, but not infected, dendritic cells express activation markers stimulated by inflammatory cytokines derived from infected cells. Third, MVA infection induces apoptosis in dendritic cells via upregulation of Bcl-2 protein NOXA[17].

Our proteomic analysis provided evidence of secretion of inflammatory cytokines and chemokines and induction of an interferon response in THP-1s infected with MVA. These macrophage responses were largely unaffected by the DNA replication blocker AraC, suggesting that incoming viral DNA genomes, rather than replication products, are sensed by innate immune sensors. Furthermore, we observed downregulation of STING following MVA infection, which strongly indicates the autophagic degradation of this sensor[41] following MVA genome detection. We did not observe upregulation of the activation marker SIGLEC1 in MVA-infected THP-1s, but this agrees with the single-cell transcriptomic analysis, which showed that only bystander, non-infected dendritic cells express activation markers[17] whereas our data represents the proteomic changes in homogeneously infected THP-1s. We did observe SIGLEC1 upregulation in THP-1s stimulated with heat-inactivated MVA, showing that this experimental model captures macrophage activation upon antigen stimulation. Last, we also observed upregulation of NOXA (PMAIP1) during MVA infection of THP-1s[17,72]. NOXA has been shown to contribute to the downregulation of the anti-apoptotic protein MCL1, whilst the anti-apoptotic protein BCL2L1 (Bcl-XL) contributed to antagonise apoptosis in MVA-infected cells and did not change in abundance[73]. Our proteomic analysis of infected THP-1s also captured known abundance behaviour of both these anti-apoptotic proteins.

Other than corroborating previous observations, our proteomic analysis also provided multiple insights into the interactions between MVA and antigen-presenting myeloid cells. The downregulation of IL-1β and the modulation of the abundance of proteins involved in inflammasome activation implicates this immune sensing pathway and inflammatory, pyroptotic cell death in the response to MVA infection in human macrophages. Our data supports that caspase-1-dependent pyroptosis is the dominant mechanism of cell death of MVA-infected macrophages. Notably, MVA lacks a functional orthologue of B13, a viral antagonist of caspase-1 but still encodes F1, an apoptosis and inflammasome inhibitor, and A47, a homologue of gasdermins that counteracts IL-1β secretion and pyroptosis[57,74,75]. Therefore, we predict that F1 and A47 are largely non-functional during MVA infection because we observed inflammasome assembly, GSDMD processing, IL-1β secretion and pyroptotic cell death in MVA-infected macrophages. MVA also lacks C1, a viral protein recently shown to enhance inflammasome activation[74]. It is likely that viral DNA triggers inflammasome activation and pyroptosis via the absent in melanoma 2 (AIM2) inflammasome[76] and/or NLRP3 inflammasome downstream of STING activation[77,78]. Inflammasome-dependent release of cytokines and antigens derived from pyroptotic cells shape adaptive antigen-specific immunity[39,79]. It remains to be determined how activation of the inflammasome and pyroptosis contribute to the immunogenicity and protection of MVA-based vaccines in vivo.

Proteins upregulated by heat-inactivated MVA included five members of the poly-ADP-ribosyl polymerase (PARP) family, some of which have well-characterised roles in antiviral responses, such as ZC3HAV1 (also known as PARP13 or ZAP)[80]. ZAP is an antiviral factor that restricts MVA infection in human cells[67]. VACV-WR induces the proteasomal degradation of ZAP and encodes at least one protein that interacts with and overcomes its antiviral function[18,67]. Blockade of MVA DNA replication and post-replicative gene expression upregulated ZAP, whilst PARPs 9, 10, 12 and 14 were only upregulated by heat-inactivated virus. This suggests that these latter PARPs might be targeted by early gene product(s), may indicate that they serve as-yet uncharacterised antiviral function, and highlights the usefulness of our multi-level approach, that included MVA infection with or without a replication inhibitor (AraC), and heat-inactivated virions, to reveal regulation that would otherwise not be observed.

In sum, our quantitative temporal analysis of the infection of human cells by the MVA vaccine has revealed an extensive remodelling of the host proteome. This highlighted that, at different stages of infection, MVA retains a limited capacity to antagonise innate immune sensing and exhibits features not present in the VACV-WR strain related to the first-generation vaccine. Our data offer an opportunity to dissect how multiple, seemingly redundant MVA and VACV immunomodulatory effectors function, to discover viral strategies to escape antiviral immunity and to explore how sensing of MVA infection shape adaptive immune responses to vaccination and immunological memory. A comprehensive understanding of all these issues may be essential to generate future, fourth-generation smallpox/mpox vaccines in addition to providing the basis for future genomic editing of MVA to better antagonise the host immune response and provide a more effective vaccine vector.

## Methods

### Cells and viruses

Primary human foetal foreskin fibroblast cells immortalised with human telomerase (HFFF-TERTs, male) and primary chicken embryo fibroblasts (CEFs) were grown in Dulbecco's modified Eagle's medium (DMEM) supplemented with foetal bovine serum (FBS: 10% v/v), and penicillin/streptomycin at 37 °C in 5% $CO_2$. HFFF-TERTs have been tested at regular intervals since isolation to confirm both that the HLA and MICA genotypes, and the morphology and antibiotic resistances are consistent with the original description[81]. In addition, human fibroblasts (dermal or foreskin) are the only permissive cells for human cytomegalovirus (HCMV) in cell culture, and HFFF-TERTs are routinely infected with the HCMV Merlin strain, further limiting the chances that the cells have been contaminated with another cell type[21]. Fresh CEFs were obtained from the Pirbright Institute (Woking, UK) and directly seeded for virus production without further passages. THP-1 monocytes (ATCC, TIB-202) and derivative cell line expressing caspase-1 CARD fused to EGFP (C1C-EGFP)[40] were grown in RPMI 1640 medium supplemented with 10% FBS, GlutaMAX (Thermo Fisher Scientific), and penicillin/streptomycin at 37 °C in 5% $CO_2$; additionally, 200 µg/ml hygromycin was added to the growth medium of THP-1[C1C-EGFP] cells. All cell lines used regularly tested negative for mycoplasma.

Modified vaccinia Ankara (MVA) was obtained as a seed stock prepared from an original MVA stock at passage 575[82]. A MVA strain expressing GFP under the control of a synthetic vaccinia early/late promoter (MVA-GFP) was described elsewhere[30]. MVA and MVA-GFP were propagated in CEFs, purified by ultracentrifugation through two 36% (w/v) sucrose cushions and suspended in 10 mM Tris-HCl pH 9.0. MVA infectivity was determined by immunocytochemistry on CEFs and HFFF-TERTs cells by using a polyclonal rabbit anti-VACV antibody[83]. For heat inactivation, MVA stock was diluted ten-fold in DMEM or RPMI 1640 supplemented with 2% FBS and heated at 56 °C for 1 h in a water bath[84,85].

## Plasmids and lentiviral transduction

The human *GLE1* (NM_001003722.2:88-2184 [https://www.ncbi.nlm.nih.gov/nuccore/NM_001003722.2/]), *ISG20* (NM_002201.6:86-631 [https://www.ncbi.nlm.nih.gov/nuccore/NM_002201.6]), *NSA2* (NM_014886.6:113-895 [https://www.ncbi.nlm.nih.gov/nuccore/NM_014886.6]), *NUP54* (NM_017426.4:24-1547 [https://www.ncbi.nlm.nih.gov/nuccore/NM_017426.4]), *NUP62* (NM_153719.4:412-1980 [https://www.ncbi.nlm.nih.gov/nuccore/NM_153719.4]), *NUP88* (NM_001320653.2:13-2286 [https://www.ncbi.nlm.nih.gov/nuccore/NM_001320653.2]), *OASL* (NM_003733.4:276-1820 [https://www.ncbi.nlm.nih.gov/nuccore/NM_003733.4]), *RBM28* (NM_018077.3:116-2395 [https://www.ncbi.nlm.nih.gov/nuccore/NM_018077.3]), and *ZNFX1* (NM_021035.3:86-5842 [https://www.ncbi.nlm.nih.gov/nuccore/NM_021035.3]), and firefly luciferase (*Luc*) open reading frames (ORFs) were synthesised by GenScript with flanking 5′ *Mlu*I and 3′ *Not*I restriction and lacking the stop codon. These were subcloned into the lentivector pJAGE1-C-FLAG cut with *Mlu*I and *Not*I, in frame with the sequence encoding a C-terminal FLAG epitope. The plasmid pJAGE1-C-FLAG is a derivative of pHAGE-SFFV[86]. The cloned lentivectors were used to generate lentiviral particles for transduction of HFFF-TERTs.

HEK 293T (ATCC CRL-11268) seeded in 12-well plates were transfected with 500 ng lentivector, and 200 ng and 400 ng of helper plasmids pVSV-G and pCMV-dR8.91, respectively, with TransIT LT1 transfection reagent (Mirus). Supernatants containing lentiviral particles were harvested 48 h post-transfection, filtered with a 0.45-µm syringe filter to remove cellular debris, and serial dilutions of the filtered supernatants were used to transduce HFFF-TERTs seeded in 12-well plates. After 48 h, transduced cells with stable, constitutive expression of the transgene were selected with 1 µg/ml puromycin for at least three passages.

## Viral infections and inhibitors

For proteomic experiments, $7.5 \times 10^5$ HFFF-TERTs were seeded in a 25-cm² flask two days prior to infection. Alternatively, $2.25 \times 10^6$ THP-1 cells were seeded in a 25-cm² flask in presence of 20 ng/ml PMA (phorbol 12-myristate 13-acetate, Cayman Chemical) for 48 h for macrophage differentiation prior to infection. Cells were infected with MVA, or the equivalent amount of heat-inactivated MVA, at MOI 5 for 2, 4, 8, 12, or 18 h for HFFF-TERTs, or 4 and 12 h for PMA-differentiated macrophages, with time zero defined as the time the virus was added to the cells. Mock-infected controls were harvested at 12 h post-infection for HFFF-TERTs or at 4 and 12 h post-infection for THP-1 macrophages. Where indicated, cells were incubated with cytosine arabinoside (AraC) at 40 µg/ml from the time zero. The time course experiments were performed in biological duplicate, using the same virus and cell stocks but performed on different days. The 12-h or the 4-h time points of mock-infected cells are shown as time zero in the graphs for HFFF-TERTs or THP-1 data, respectively.

For the gain-of-function screen, overexpression cell lines were seeded in duplicate in 24-well plates at $1.35 \times 10^5$ cells per well one day prior to infection, or $7.0 \times 10^5$ cells per well two days prior to infection. Cells were infected with MVA-GFP at MOI 0.01 for two days.

For immunoblotting, $4.5 \times 10^5$ overexpression cell lines were seeded in 6-well dishes for one day, or $1.0 \times 10^6$ THP-1 cells were seeded 6-well plates in presence of 20 ng/ml PMA for two days. Then, THP-1 macrophages were treated with 40 µM VX-765 (Cayman Chemical), 25 µM z-VAD-fmk (BD Pharmingen) or the equivalent amount of DMSO, for 30 min and infected with MVA at MOI 5 for 8 h in the continued presence of VX-765 or DMSO.

For the LDH release assay, $3.0 \times 10^4$ THP-1s were seeded in triplicate in 96-well plates in presence of 20 ng/ml PMA and, two days later, infected with MVA at MOI 5 for 8 h. The infection was carried out in presence of the following inhibitors, which were added 30 min before infection: 0.5 µM necrosulfonamide (NSA, Merck), 30 µM necrostatin-1 (Cayman Chemical), 1.5 µM GSK'872 (Cayman Chemical), 40 µM VX-765 (Cayman Chemical), 25 µM z-VAD-fmk (BD Pharmingen) or the equivalent volume of DMSO. As control for the activity of necroptosis inhibitors, HFFF-TERTs were seeded in 96-well plate ($1.5 \times 10^4$ cell/well) and next day, necroptosis was stimulated with 30 ng/ml TNFα (R&D systems), 5 µM BV-6 (Selleckchem), and 25 µM z-VAD-fmk in presence of DMSO, NSA, necrostatin-1, or GSK'872 for 24 h.

For fluorescence microscopy, $4.5 \times 10^5$ THP-1 or THP-1^CIC-EGFP cells seeded on 13-mm coverslips in 12-well plates were treated with 40 µM VX-765 (Cayman Chemical) for 30 min and then, infected with MVA at MOI 5 for 4 and 8 h in the continued presence of VX-765.

## Flow cytometry

MVA-infected HFFF-TERTs or PMA-differentiated THP-1 macrophages were detached with trypsin-EDTA (Gibco) 24 h post-infection, passed through a 70-µm cell strainer, washed in PBS and stained with Zombie Violet viability dye (BioLegend) for 30 min at 4 °C in the dark. Cells were collected by centrifugation, washed once with PBS and fixed in Cytofix/Cytoperm fixation and permeabilization solution (BD Biosciences) for 30 min at 4 °C in the dark. Cells were washed twice with Perm/Wash buffer (BD Biosciences) and stained with a polyclonal rabbit anti-VACV antibody[83] diluted 1:500 in Perm/Wash buffer, followed by AlexaFluor 568-conjugated goat anti-rabbit IgG (Invitrogen) diluted 1:100. Alternatively, transduced HFFF-TERTs were harvested as described above and stained with rabbit anti-FLAG (Cell Signalling Technology, #2368) diluted 1:200 and AlexaFluor 488-conjugated goat anti-rabbit IgG (Abcam) diluted 1:100. Stained cells were suspended in PBS before data acquisition with an Attune NxT flow cytometer. Data were analysed with FlowJo software.

## Gain-of-function screen

GFP fluorescence and phase contrast images of MVA-GFP-infected cells were acquired every 3 h on Incucyte S3 live cell imaging instrument (Sartorius) starting at 3 h p.i. A total of 16 images were acquired per well at 10x magnification. The following parameters were applied to calculate the total GFP area per well on Incucyte Basic Analysis Software (Sartorius): background subtraction by Top-Hat segmentation with radius of 100 µm, threshold of 1.0 green calibrated unit (GCU) and edge sensitivity of −15.

## Immunoblotting

Cells were washed twice with PBS and lysed on ice with cell lysis buffer (50 mM Tris-HCl pH 8.0, 150 mM NaCl, 1 mM EDTA, 10% (v/v) glycerol, 1% (v/v) Triton X-100 and 0.05% (v/v) Nonidet P-40 (NP-40)), supplemented with protease inhibitors (cOmplete Mini, Roche), for 20 min. Lysates were transferred to fresh tubes and spun at 17,000 g for 10 min at 4 °C. Protein concentration was determined using a bicinchoninic acid protein assay kit (Pierce). After mixing with 5× SDS-gel loading buffer and boiling at 100 °C for 5 min, equivalent amounts of protein samples (15–20 µg per well) were loaded onto 4–15% Mini-PROTEAN TGX precast protein gels (Bio-Rad), separated by electrophoresis and transferred onto nitrocellulose membranes (GE Healthcare). Membranes were blocked at room temperature with 5% (w/v) non-fat milk in Tris-buffered saline (TBS) containing 0.1% (v/v) Tween-20. The membranes were incubated with specific primary antibodies diluted in blocking buffer or 5% (w/v) BSA (Fisher Scientific) at 4 °C overnight. After washing, membranes were probed with fluorophore-conjugated secondary antibodies (LI-COR Biosciences) diluted in 5% (w/v) non-fat milk at room temperature for 1 h. After washing, membranes were imaged using the Odyssey CLx imaging system (LI-COR Biosciences). The band intensities on the immunoblots were quantified using the Image Studio Lite software (LI-COR Biosciences). The antibodies used for immunoblotting are listed: rabbit anti-FLAG (Cell Signalling Technology, #2368, diluted 1:1000), rabbit anti-CANX (LSBio, LS-B6881, 1:1000), mouse monoclonal anti-GAPDH 6C5 (Millipore, MAB374, 1:2500), rabbit monoclonal anti-GSDMD E504N (Cell Signalling

Technology, #69469, diluted 1:1000), IRDye 800CW-conjugated goat anti-rabbit IgG (LI-COR, 926-32211, diluted 1:10,000), and IRDye 680LT-conjugated goat anti-mouse IgG (LI-COR, 926-68020, diluted 1:10,000).

### LDH release assay

Lactate dehydrogenase (LDH) release was measured using the Cyto-Tox 96 Non-Radioactive Cytotoxicity Assay (Promega) according to the manufacturer's instructions. To obtain the maximum LDH release, extra wells with DMSO-treated, mock-infected cells were lysed with lysis buffer (Promega) for 45 min at 37 °C prior to assay. Average absorbance values derived from culture medium alone were subtracted from each absorbance value from experimental wells, and the percentage of LDH release (i.e., cell death) was calculated relative to the LDH released from lysed cells.

### Fluorescence microscopy

Cells were fixed with fixation solution (4% paraformaldehyde, BioLegend) for 15 min at room temperature. After fixation, cells were washed three times with PBS and permeabilized with 0.5% Triton X-100 for 10 minutes. After permeabilization, cells was stained with rabbit plyclonal anti-ASC (AL177, AdipoGen, diluted 1:200) followed by Alexa-Fluor 568 goat anti-rabbit IgG (Invitrogen, diluted 1:1000). Cells was mounted with ProLong Gold Antifade with DAPI (Invitrogen). Imaging was performed using an AxioObserverZ2 (Carl Zeiss).

Inflammasome speck quantification was performed using the Imaris software (Oxford Instruments). Images were taken at 20x magnification with 3 × 3 tiles stitched together to achieve 1.9 mm × 1.9 mm area containing an average of ~3000 cells. The cell number was quantified using the DAPI channel and a particle diameter of about 6.5 μm. The inflammasome assembly was quantified using either eGFP channel (THP-1$^{CIC-EGFP}$) or red channel (THP-1) and a particle diameter of about 1.5 μm.

### Whole cell lysate protein digestion

Two independent biological replicates were performed, 'WCL1' and 'WCL2'. For each replicate, cells were washed twice with PBS, and 250 μl lysis buffer added (6 M Guanidine, 50 mM HEPES pH 8.5). Cell lifters (Corning) were used to scrape cells in lysis buffer, which was removed to an eppendorf tube, vortexed extensively then sonicated. Cell debris was removed by centrifugation at 21,000 g for 10 min twice. For half of each sample (estimated from our previous experience with HFFF-TERT cultures as ~500 μg), dithiothreitol (DTT) was added to a final concentration of 5 mM and samples were incubated for 20 min. Cysteines were alkylated with 14 mM iodoacetamide and incubated 20 min at room temperature in the dark. Excess iodoacetamide was quenched with DTT for 15 min. Samples were diluted with 200 mM HEPES pH 8.5 to 1.5 M Guanidine followed by digestion at room temperature for 3 h with LysC protease at a 1:100 protease-to-protein ratio. Samples were further diluted with 200 mM HEPES pH 8.5 to 0.5 M Guanidine. Trypsin was then added at a 1:100 protease-to-protein ratio followed by overnight incubation at 37 °C. The reaction was quenched with 5% formic acid, then centrifuged at 21,000 g for 10 min to remove undigested protein. Peptides were subjected to C18 solid-phase extraction (SPE, Sep-Pak, Waters) and vacuum-centrifuged to near-dryness.

### Peptide labelling with tandem mass tags

In preparation for TMT labelling, desalted peptides were dissolved in 200 mM HEPES pH 8.5. Peptide concentration was measured by microBCA (Pierce), and 25 μg of peptide labelled with TMT reagent. TMT reagents (0.5 mg) were dissolved in 43 μl anhydrous acetonitrile and 5 μl added to peptide at a final acetonitrile concentration of 30% (v/v). HFFF-TERTs sample labelling was: 126, MVA (2 h infection); 127 N, MVA+AraC (2 h); 127 C, MVA-HI (2 h); 128 N, MVA (4 h); 128 C, MVA

+AraC (4 h); 129 N, MVA-HI (4 h); 129 C, MVA (8 h); 130 N, MVA+AraC (8 h); 130 C, MVA-HI (8 h); 131 N, MVA (12 h); 131 C, MVA+AraC (12 h); 132 N, MVA-HI (12 h); 132 C, MVA (18 h); 133 N, MVA+AraC (18 h); 133 C, MVA-HI (18 h); 134 N, mock infection. THP-1 sample labelling was: 126b, Mock (4 h) WCL1; 127 N, MVA (4 h) WCL1; 127 C, MVA+AraC (4 h) WCL1; 128 N, MVA-HI (4 h) WCL1; 128 C, Mock (12 h) WCL1; 129 N, MVA (12 h) WCL1; 129 C, MVA+AraC (12 h) WCL1; 130 N, MVA-HI (12 h) WCL1; 130 C, Mock (4 h) WCL2; 131 N, MVA (4 h) WCL2; 131 C, MVA+AraC (4 h) WCL2; 132 N, MVA-HI (4 h) WCL2; 132 C, Mock (12 h) WCL2; 133 N, MVA (12 h) WCL2; 133 C, MVA+AraC (12 h) WCL2; 134 N, MVA-HI (12 h) WCL2. TMT impurities from both experiments are detailed in Supplementary Data 6. Following incubation at room temperature for 1 h, the reaction was quenched with hydroxylamine to a final concentration of 0.3% (v/v). TMT-labelled samples were combined at a 1:1:1:1:1:1:1:1:1:1:1:1:1:1:1:1 ratio. The sample was vacuum-centrifuged to near dryness and subjected to C18 SPE (Sep-Pak, Waters). An unfractionated singleshot was analysed initially to ensure similar peptide loading across each TMT channel, thus avoiding the need for excessive electronic normalisation. As all normalisation factors were >0.5 and <2 (HFFF-TERTs experiment 1: between 0.86–1.15; HFFF-TERTs experiment 2: between 0.80–1.03; THP-1 experiment: 0.98–1.57), data for each singleshot experiment was analysed with data for the corresponding fractions to increase the overall number of peptides quantified. Normalisation is discussed in 'Data Analysis', and high pH reversed-phase (HpRP) fractionation is discussed below. TMT labelling efficiency was assessed using the singleshot sample: 97.5% (HFFF-TERTs experiment 1); 99.7% (HFFF-TERTs experiment 2; THP-1 experiment: 98.1%).

### Offline HpRP fractionation

TMT-labelled tryptic peptides were subjected to HpRP fractionation using an Ultimate 3000 RSLC UHPLC system (Thermo Fisher Scientific) equipped with a 2.1 mm internal diameter (ID) x 25 cm long, 1.7 μm particle Kinetix Evo C18 column (Phenomenex). Mobile phase consisted of A: 3% acetonitrile (MeCN), B: MeCN and C: 200 mM ammonium formate pH 10. Isocratic conditions were 90% A / 10% C, and C was maintained at 10% throughout the gradient elution. Separations were conducted at 45 °C. Samples were loaded at 200 μl/min for 5 min. The flow rate was then increased to 400 μl/min over 5 min, after which the gradient elution proceed as follows: 0–19% B over 10 min, 19–34% B over 14.25 min, 34–50% B over 8.75 min, followed by a 10 min wash at 90% B. UV absorbance was monitored at 280 nm and 15 s fractions were collected into 96-well microplates using the integrated fraction collector. Fractions were recombined orthogonally in a checkerboard fashion, combining alternate wells from each column of the plate into a single fraction, and commencing combination of adjacent fractions in alternating rows. Wells were excluded prior to the start or after the cessation of elution of peptide-rich fractions, as identified from the UV trace. This yielded two sets of 12 combined fractions, A and B, which were dried in a vacuum centrifuge and resuspended in 10 μl MS solvent (4% MeCN / 5% formic acid) prior to LC-MS3. 12 set 'A' fractions were used for MS3 analysis of all experiments, and half of each sample was subjected to Synchronous Precursor Selection MS3 (SPS-MS3) analysis, with the other half subjected to real-time search (RTS) MS3 analysis[87] (described below).

### LC-MS3 for TMT and TMT/SILAC experiments

For 'singleshot' analysis of unfractionated peptides from each experiment, mass spectrometry data was acquired using an Orbitrap Lumos. Subsequently, fractions were acquired using an Orbitrap Eclipse.

For Orbitrap Lumos analyses, an Ultimate 3000 RSLC nano UHPLC equipped with a 300 μm ID x5 mm Acclaim PepMap μ-Precolumn (Thermo Fisher Scientific) and a 75 μm ID x50 cm 2.1 μm particle Acclaim PepMap RSLC analytical column was used. Loading solvent was 0.1% FA, analytical solvent A: 0.1% FA and B: 80% MeCN +0.1% FA. All separations were carried out at 40 °C. Samples were

loaded at 5 μL/min for 5 min in loading solvent before beginning the analytical gradient. The following gradient was used: 3–7% B over 2 min, 7–37% B over 173 min, followed by a 4 min wash at 95% B and equilibration at 3% B for 15 min. Each analysis used a MultiNotch MS3-based TMT method[88]. The following settings were used: MS1: 380–1500 Th, 120,000 resolution, 2 ×10^5 automatic gain control (AGC) target, 50 ms maximum injection time. MS2: Quadrupole isolation at an isolation width of m/z 0.7, collision-induced dissociation (CID) fragmentation (normalised collision energy (NCE) 34) with ion trap scanning in turbo mode, with 1.5 ×10^4 AGC target and 120 ms maximum injection time. MS3: In SPS mode the top 10 MS2 ions were selected for higher-energy collisional dissociation (HCD) fragmentation (NCE 45) and scanned in the Orbitrap at 60,000 resolution with an AGC target of 1×10^5 and a maximum accumulation time of 150 ms. Ions were not accumulated for all parallelisable time. The entire MS/MS/MS cycle had a target time of 3 s. Data analysis is discussed below.

For Orbitrap Eclipse analyses, the FAIMS Pro interface (Thermo Fisher Scientific) was coupled to a Proxeon EASY-nLC 1200 liquid chromatograph (LC) (ThermoFisher Scientific). Peptides were separated on a 100 μm inner diameter microcapillary column packed with ~35 cm of Accucore150 resin (2.6 μm, 150 Å, ThermoFisher Scientific). For each analysis, 1–2 μg of peptide was loaded onto the column and fractionated over a 90 min gradient of 7 to 27% acetonitrile in 0.125% formic acid at a flow rate of ~600 nL/min. Mass spectrometric data for each fraction of each sample were collected using two distinct data acquisition modes (SPS-MS3 and RTS-MS3), all with FAIMS. SPS-MS3 data were collected with a FAIMS compensation voltage (CV) set of −40 V, −60 V, and −80 V, while RTS-MS3 data were collected with a FAIMS CV set of −30 V, −50 V, and −70 V, with each segment as a 1 s TopSpeed method. For both acquisition methods, the scan sequence began with an MS1 spectrum (Orbitrap analysis; resolution, 60,000; mass range, 350–1350 Th; automatic gain control (AGC) target 100%; maximum injection time, auto). Precursors (charge states 2–5; precursor fit 50% at 0.7 Th; minimum intensity of 15 K) were then selected for MS2/MS3 analysis[89]. MS2 analysis consisted of collision-induced dissociation (CID) with quadrupole ion trap analysis, using the following parameters: scan speed, turbo; AGC target, 100%; NCE, 35; q-value, 0.25; maximum injection time, 35 ms; and isolation window, 0.5 Th. MS3 precursors were fragmented by HCD and analysed using the Orbitrap with the following parameters: resolution, 50,000; NCE, 55; AGC, 250%; maximum injection time, 86 ms; maximum synchronous precursor selection (SPS) ions, 10; and isolation window, 1.2 Th. RTS-MS3 data were collected with the following parameters: "close-out" 2; peptide length, 7; maximum number of SPS Ions, 10; minimum SPS m/z, 378 Th; AGC target, 250,000; source RF, 30%; MS2 isolation width, 1.2 Th; MS3 isolation width, 3 Th; MS2 collision energy, 35%; MS3 collision energy, 55%; resolution, 50,000; maximum injection time, 200 s:; search tolerance ±10ppm; minimum XCorr, 0.5; minimal delta Cn 0.05. Dynamic exclusion was set at 120 s with ±10 ppm error tolerance.

### Data analysis

In the following description, the first report in the literature for each relevant algorithm is listed. Mass spectra were processed using a Sequest-based software pipeline for quantitative proteomics developed by Professor Steven Gygi's laboratory at Harvard Medical School. MS spectra were converted to mzXML using an extractor built upon Thermo Fisher's RAW File Reader library (version 4.0.26).

After converting RAW files to mzxml format, MS data were analysed to correct errors in monoisotopic peak assignment using a published algorithm[90] and then searched against a database of protein sequences. A combined database was constructed from (a) the human Uniprot database (accessed 20th June 2022, UP000005640), (b) the MVA proteome (accessed 20th June 2022, UP000172909), (c) common contaminants such as porcine trypsin and endoproteinase LysC. The combined database was concatenated with a reverse database composed of all protein sequences in reversed order. Sequest searches were performed using a 20 ppm precursor ion tolerance[91,92]. Product ion tolerance was set to 1 Da. Oxidation of methionine residues (15.99492 Da) was set as a variable modification. TMT tags on lysine residues and peptide N termini (304.207145 Da) and carbamidomethylation of cysteine residues (57.02146 Da) were included as static modifications. Peptides were assumed to be fully tryptic with up to two missed cleavages.

To control the fraction of erroneous protein identifications, a target-decoy strategy was employed[93]. Peptide spectral matches (PSMs) were filtered to an initial peptide-level false discovery rate (FDR) of 1% with subsequent filtering to attain a final protein-level FDR of 1%. PSM filtering was performed using a linear discriminant analysis, as described previously[93]. This distinguishes correct from incorrect peptide IDs in a manner analogous to the widely used Percolator algorithm (https://noble.gs.washington.edu/proj/percolator/), though employing a distinct machine learning algorithm. The following parameters were considered: XCorr, ΔCn, missed cleavages, peptide length, charge state, and precursor mass accuracy. Peptides shorter than seven amino acids in length or with XCorr less than 1.0 were excluded prior to LDA filtering. Protein assembly was guided by principles of parsimony to produce the smallest set of proteins necessary to account for all observed peptides (algorithm described previously[93]).

Proteins were quantified by summing TMT reporter ion counts across all matching peptide-spectral matches, as described previously[88]. Briefly, a 0.003 Th window around the theoretical m/z of each reporter ion (126, 127n, 127c, 128n, 128c, 129n, 129c, 130n, 130c, 131n, 131c, 132n, 132c, 133n, 133c, 134n) was scanned for ions, and the maximum intensity nearest to the theoretical m/z was used. An isolation specificity filter with a cutoff of 50% was employed to minimise peptide co-isolation[88]. Peptide-spectral matches with poor quality MS3 spectra (more than 15 TMT channels missing and/or a combined signal-to-noise (S:N) ratio of less than 250 across all TMT reporter ions) or no MS3 spectra at all were excluded from quantitation. Peptides meeting the stated criteria for reliable quantitation were then summed by parent protein, in effect weighting the contributions of individual peptides to the total protein signal based on their individual TMT reporter ion yields. In order to maximise coverage, the dataset was not filtered to eliminate proteins with single peptide quantitations. Protein quantitation values were exported for further analysis in Excel.

For protein quantitation, reverse and contaminant proteins were removed, then each reporter ion channel was summed across all quantified proteins and normalised assuming equal protein loading across all channels. For further analysis and display, fractional TMT signals were used (i.e. reporting the fraction of maximal signal observed for each protein in each TMT channel, rather than the absolute normalised signal intensity). In three instances (for MVA proteins MVA045 and MVA131 and human protein DZIP1L), the S:N in the mock sample was zero. For the purposes of calculating fold change of each infected sample compared to mock, these mock values were set to 2% of the maximum S:N for any sample in the same WCL experiment. Normalised spectral abundance factors were calculated for a given protein from the sum of peptides quantified divided by the sequence length, then corrected to the sum of all abundance factors. Separate abundance factors were calculated for each cell type.

For analysis of the viral proteome, the MVA proteome (UP000172909) was matched with the VACV-WR proteome (UP000000344) using the PathoSystems Resource Integration Center (PATRIC)'s proteome comparison tool with default settings[94].

Hierarchical centroid clustering based on uncentered Pearson correlation, and k-means clustering were performed using Cluster 3.0 (Stanford University) and visualised using Java Treeview (http://jtreeview.sourceforge.net) unless otherwise noted.

Throughout the manuscript, data are presented using the Excel 'XY scatter with smoothed lines function'. This uses a Bezier curve interpolation to represent the trend. For accuracy, markers of individual timepoints are also shown.

### Statistical analysis and reproducibility

For the proteomic experiments, two biological replicates were employed throughout. P-values were estimated using the methods of Significance A or B as implemented in Perseus version 1.5.1.6[95,96]. A one-way ANOVA test also implemented in Perseus version 1.5.1.6 was used to identify proteins differentially expressed over time in MVA-infected samples compared to mock, MVA+AraC samples compared to mock and MVA-HI samples compared to mock. P-values were corrected using the method of Benjamini–Hochberg to control for multiple testing error. The Fisher's exact test was calculated using GraphPad Prism 9. No statistical method was used to predetermine sample size. No data were excluded from the analyses, except for Fig. 2e; images containing bright fluorescent artifacts or rare MVA-GFP plaques[97] were excluded from analysis because they skewed the calculation of the GFP area. The experiments were not randomised and the investigators were not blinded to allocation during experiments and outcome assessment.

For Fig. 2e, we analysed four independent gain-of-function experiments, each carried in duplicate. P-values were estimated using a two-way ANOVA test and post-hoc Dunnett's multiple comparisons test using GraphPad Prism 9.

For Fig. 4a, we analysed three independent experiments and for 4c, two independent experiments, each carried out in duplicate. P-values were estimated by one-way ANOVA and post-hoc Dunnett's multiple comparisons test using GraphPad Prism 9. For Fig. 4e, we analysed two independent experiments.

### Pathway analysis

The Database for Annotation, Visualisation and Integrated Discovery (DAVID) was used to determine pathway enrichment[24]. A given cluster was always searched against a background of all proteins quantified within the relevant experiment. Downregulated proteins were defined as those factors that decreased >2-fold in abundance compared to the mock sample at ≥1 time point during infection. Proteins that were not downregulated were defined as factors that decreased <1.25-fold in abundance compared to the mock sample at all times of infection. Upregulated proteins were defined in a similar manner.

### Interaction analysis

Interaction analysis was performed on groups of proteins defined by changes in abundance during MVA, MVA+AraC, MVA-HI and VACV infection of HFFF-TERTs indicated in Figs. 2c and 3b. Network analyses were performed using interactions from BioPlex 3.0[29], whereby networks from 293T and HCT116 cells were merged to act as a comparator to our present data. All analyses were performed in Mathematica 13.1 (Wolfram Research).

To assess the tendency for sets of differentially expressed proteins to interact with each other, each was mapped onto the combined BioPlex network after converting all protein identifiers to Entrez GeneIDs. Graph assortativity was then calculated and the process repeated using 1000 randomised protein sets of equal size. Scores from randomised protein sets were then fit to a Gaussian distribution and Z-scores and p-values were estimated.

To produce interaction networks shown in Fig. 2d and Supplementary Fig. 4d, proteins were grouped according to their differential expression under each experimental condition. Each group was then mapped onto the BioPlex PPI network analysis and network propagation was used to identify additional nodes that closely associate with proteins in each group. Network propagation was calculated via random walk with restart as described previously[98], with the restart

probability set to 0.5. After 40 iterations, proteins were sorted by descending weights. In Fig. 2e the top 100 highest ranking proteins were selected for display. In Supplementary Fig. 4d the top 150 highest ranking proteins were selected for display, though for clarity those proteins that did not interact with any other proteins on this list were excluded.

### Reporting summary

Further information on research design is available in the Nature Portfolio Reporting Summary linked to this article.

## Data availability

The proteomic data generated in this study have been deposited in the ProteomeXchange Consortium via the PRIDE[99] partner repository under accession code PXD039034. Furthermore, all peptides quantified in this study are provided in Supplementary Data 5. All materials described in this manuscript, and any further details of protocols employed can be obtained on request from the corresponding authors by email to jonas.albarnaz@pirbright.ac.uk or mpw1001@cam.ac.uk. Source data are provided with this paper, including the data for all graphs.

## Code availability

The data analysis pipeline as a whole may be licensed from Harvard Medical School, and constituent components have been published and referenced above. These include a RAW file to mzXML conversion tool, built upon Thermo Fisher's RAW File Reader library (version 4.0.26); a monoisotopic mass assignment tool[90]; the Sequest algorithm[91]; strategies for target-decoy analysis, peptide spectral match filtering and protein assembly[93], and methods for protein quantification and isolation specificity filtering[88] are recapitulated in Proteome Discoverer Software (Thermo Fisher, UK). For licensing enquiries, contact Steven Gygi (steven_gygi@hms.harvard.edu).

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

## Acknowledgements

We thank R. Blasco (INIA-CSIC, Spain) and C. Maluquer de Motes (University of Surrey, UK) for providing the MVA-GFP strain. This research was funded by the UKRI Mpox Research Consortium, grant number BB/X011143/1. Funding also derived from Medical Research Council Project Grant MR/W025647/1 to M.P.W., R01 GM132129 to J.A.P., R021 GM67945 to S.P.G. and National Institute of Health grant U24HG006673 to E.L.H and S.P.G. This study was additionally supported by the Cambridge Biomedical Research Centre, UK.

## Author contributions

Conceptualisation: M.P.W. Investigation: J.D.A., M.O., J.K., H.L., Y.D., J.A.P., R.A. Data analysis: J.D.A., M.O., J.K., H.L., Y.D., E.L.H., M.P.W. Funding acquisition: S.P.G, E.L.H, G.L.S, M.P.W. Supervision: S.P.G, E.L.H, G.L.S, M.P.W. Writing: J.D.A., M.O., J.K., J.A.P., R.A., S.P.G, E.L.H, G.L.S, M.P.W.

## Competing interests

The authors declare no competing interests.
