## [Peer Review File · Nature Communications]

Reviewers' Comments:

Reviewer #2:

Remarks to the Author:

In the revision of Quantitative temporal proteomic analysis of modified vaccinia Ankara, a monkeypox virus vaccine by Albarnaza et al., the authors provide address many of the concerns of the reviewers. I continue to believe that the dataset is novel, important, and likely be a critical resource for the community. However, the study remains descriptive, and the lack of orthogonal validation and demonstration of therapeutic relevance continues to hamper the overall potential impact of this study

Major points:

1) The authors performed quantitative temporal viromics (QTV) of modified vaccinia Ankara (MVA) whose variant strain is used for as a vaccine against smallpox and monkeypox. They found nucleoporins as one of the key elements for MVA antagonism. Investigation of the mechanism and functional consequences of the downregulation of those factors would be critical and the data would improve their manuscript. The study was done in immortalized cell lines, and thus the therapeutic/physiological/in vivo importance of the findings is not clear. Providing functional validation of at least a subset of the dataset would provide confidence that these findings are relevant to vaccine biology and thus worth the time and attention of the community to follow up on.

2) I appreciate the authors repeated the proteomic analysis in PMA-differentiated THP-1 macrophage-like cells. The manuscript will greatly benefit from a more comprehensive analysis comparing the data generated in HFFF-TERTs and THP-1 to elucidate potential common and cell type-specific virus-regulated proteins.

3) While they analyzed similarities and difference carefully between the cell types as well as previous similar studies, it seems to end in a list of their observation. The discussion lacks significance of factors they found from the perspective of vaccine design and development.

4) Systematic errors and incorrect peptide assignment increasing the false peptide identification can occur in high-throughput analysis due MS/MS fragmentation patterns. The authors suggested to maximize coverage, the dataset was not filtered to eliminate proteins with single peptide quantification. Since this manuscript is the first global view of the impacts of MVA infection on the host proteome, and low number of biological replicates, to avoid false positive rates, more conservative criteria should be applied ≥ 2 unique peptides to be identified within a single protein for its positive identification.

Minor points:

1) TMT reagents have different isotope impurities that need to be include for database search to correct the reporter ion ratio interference across different TMT channels.

Reviewer #6:

Remarks to the Author:

In this manuscript, the authors conducted a multiplexed proteomic analysis of MVA and host at five time points throughout MVA infection of human cells. The experiment included inactivated controls to reveal the contribution of the viral particle with no-, or limited viral gene expression, and might provide a global view of the impact of MVA infection on the host proteome.

As for the major points mentioned by reviewer #4 that the low number of biological replicates were used, a further proteomic analysis in differentiated THP-1 cells was performed in revision, and differences in regulation between cell types were discussed. As for the minor points mentioned by reviewer #4, appropriate modifications have been made in this revision. Furthermore, the meaning of the dots color should be presented clearly in the legends of Figure 2d and 4d.

REVIEWER COMMENTS

Reviewer #2 (Remarks to the Author):

In the revision of Quantitative temporal proteomic analysis of modified vaccinia Ankara, a monkeypox virus vaccine by Albarnaza et al., the authors provide address many of the concerns of the reviewers. I continue to believe that the dataset is novel, important, and likely be a critical resource for the community. However, the study remains descriptive, and the lack of orthogonal validation and demonstration of therapeutic relevance continues to hamper the overall potential impact of this study

Thank you for the positive evaluation of our revised manuscript. We have now addressed the lack of orthogonal validation in two different ways.

Firstly, we have included a gain-of-function screen of (i) selected proteins that were down-regulated during MVA infection of HFFF-TERTs and (ii) selected proteins whose induction was suppressed in comparison to infection in presence of DNA replication inhibitor or heat-killed virions. This revealed that one interferon-inducible effector, ISG20, can limit viral gene expression (new **Figure 2e**, **Extended Data Figure 4**). We discuss this finding in the context of the well-characterised capacity of MVA vectors to attain high levels of heterologous gene expression under the control viral promoters:

Results

“To gauge the functional significance of the host proteomic changes during MVA infection, we performed a small-scale gain-of-function screen via ectopic expression in HFFF-TERTs (Extended data Fig. 4a, b). We included a selection of proteins that were downregulated during MVA infection (GLE1, NSA2, NUP54, NUP62, NUP88, RBM28). We also examined IFN-stimulated proteins whose induction was suppressed by infection, but not by heat-inactivated virions or upon inhibition of viral DNA replication (ISG20, OASL, ZNFX1). Cells were infected with an MVA strain expressing GFP under the control a viral early/late promoter (MVA-GFP)³⁰ as proxy of viral gene expression. Out of the nine selected proteins, only ISG20 significantly altered MVA-GFP infection (Fig. 2e). Of note, it is possible that other proteins expressed in this screen at a level lower than ISG20 including GLE1, NSA2 and ZNFX1 might nevertheless exhibit an effect (Extended data Fig. 4a, b).”

Discussion

“Suppression of the IFN response might represent one of the mechanisms responsible for the capacity of MVA vectors to attain high levels of heterologous gene expression under the control of viral promoters⁵⁰. Of three ISGs included in our gain-of-function screen, we could confidently conclude that ectopic ISG20 can limit MVA gene expression whereas ectopic OASL cannot. A 3'-5' exonuclease with strong preference for RNA, ISG20 can also degrade deaminated viral DNA (reviewed in ⁵⁸). Genomic analyses support that APOBEC3-mediated deamination of the viral genome is a key mutational driver in the MPXV strain that emerged globally in 2022¹. A plausible hypothesis is that the degradation APOBEC3-deaminated viral DNA by ISG20 might constitute another host defence against poxviruses. In light of the recent proposal that subset of "dominant" ISGs may confer most of the inhibition of a given virus by IFNs⁵⁹, it is not surprising that, out of three ISGs, only ISG20 exerted significant antiviral activity against MVA.”

Secondly, we followed up the observation that IL-1 β is down-regulated from whole cell lysates following infection of THP-1 macrophages, an indication that pyroptosis is induced by MVA. We confirmed this hypothesis with three different approaches: (i) lactate dehydrogenase (LDH) release assay in presence of different inhibitors of necroptosis, pyroptosis and apoptosis; (ii) immunoblotting of gasdermin D (GSDMD), the pore-forming effector of pyroptotic cell death; and (iii) fluorescence microscopy of a reporter cell line expressing caspase-1 CARD fused to GFP and immunostaining of ASC, the inflammasome adapter protein (new **Figure 4, Extended Data Figure 8**). Combined, these approaches confirmed that MVA infection of THP-1 macrophages activates the inflammasome which leads to caspase-1-dependent processing of GSDMD and cell death. We discuss these findings in the context of vaccine immunogenicity, since inflammasome-dependent release of cytokines and antigens derived from pyroptotic cells shape adaptive antigen-specific immunity. However, demonstration of the importance of inflammasome and pyroptosis to the immunogenicity of MVA-based vaccines *in vivo* would require extensive experimentation in mouse models and is beyond the scope of this study.

Results

“Inflammasomes are multimeric protein complexes that assemble in response to infectious or cellular damage stimuli and activate pro-inflammatory caspases, most notably caspase-1. Active caspase-1 cleaves pro-IL-1 β , pro-IL-18, and gasdermin D (GSDMD) (Deets & Vance 2021). To determine whether inflammasomes are indeed activated during MVA infection of THP-1s, we first measured cell death by lactate dehydrogenase (LDH) release assay. Infection of THP-1s with MVA induced a sharp increase in cell death, which was reversed by treatment with a caspase-1 inhibitor (VX-765) and, to a lesser extent, the pan-caspase inhibitor (zVAD-fmk) (Fig. 4a). Conversely, inhibitors of the necroptosis mediators RIPK1, RIPK3, and MLKL did not affect MVA-induced cell death at concentrations that blocked necroptosis in HFFF-TERTs (Fig. 4a, Extended data Fig. 8a). Next, we sought to visualize and quantify inflammasome assembly in infected THP-1 macrophages engineered to express caspase-1 caspase recruitment domain (CARD) fused to EGFP (C1C-EGFP)⁴⁰. Following infection with MVA in the presence of VX-765 to avoid cell loss due to pyroptosis, C1C-GFP redistributed to single specks in each cell at 4 and 8 h p.i. (Fig. 4b, c; Extended data Fig. 8a). Co-localisation with the adaptor protein apoptosis-associated speck-like protein containing a CARD (ASC) confirmed the identity of the specks as inflammasomes (Fig. 4b; Extended data Fig. 8a). ASC specks were also observed following infection of parental THP-1s with MVA (Extended Data Fig 8b, c). Lastly, we also observed cleavage of GSDMD downstream of inflammasome activation by MVA infection. VX-765, as well as zVAD-fmk, prevented GSDMD cleavage, indicating its caspase-1-dependence (Fig. 4d, e).”

Discussion

“Other than corroborating previous observations, our proteomic analysis also provided multiple novel insights into the interactions between MVA and antigen-presenting myeloid cells. The downregulation of IL-1 β and the modulation of the abundance of proteins involved in NLRP3 inflammasome activation implicates this immune sensing pathway and inflammatory, pyroptotic cell death in detection of and response to MVA infection in human macrophages. Our data supports that caspase-1-dependent pyroptosis is the dominant mechanism of cell death of MVA-infected macrophages. Notably, MVA lacks a functional orthologue of B13, a viral antagonist of caspase-1 but still encodes F1, an apoptosis and inflammasome inhibitor, and A47, a homologue of gasdermins that counteracts IL-1 β secretion and pyroptosis^{57,74,75}. Therefore, we predict that F1 and A47 are largely non-functional during MVA infection because we observed inflammasome assembly, GSDMD processing, IL-1 β secretion and pyroptotic cell death in MVA-infected macrophages. MVA also lacks C1, a viral protein recently shown to enhance inflammasome activation⁷⁴. It is likely that viral DNA triggers inflammasome activation and pyroptosis via the absent in melanoma 2 (AIM2) inflammasome⁷⁶ and/or NLRP3 inflammasome

downstream of STING activation^{77,78}. Inflammasome-dependent release of cytokines and antigens derived from pyroptotic cells shape adaptive antigen-specific immunity^{39,79}. It remains to be determined how activation of the inflammasome and pyroptosis contribute to the immunogenicity and protection of MVA-based vaccines in vivo.”

We have additionally edited both the abstract and the title to reflect this substantial new data.

Major points:

1) The authors performed quantitative temporal viromics (QTV) of modified vaccinia Ankara (MVA) whose variant strain is used for as a vaccine against smallpox and monkeypox. They found nucleoporins as one of the key elements for MVA antagonism. Investigation of the mechanism and functional consequences of the downregulation of those factors would be critical and the data would improve their manuscript. The study was done in immortalized cell lines, and thus the therapeutic/physiological/in vivo importance of the findings is not clear. Providing functional validation of at least a subset of the dataset would provide confidence that these findings are relevant to vaccine biology and thus worth the time and attention of the community to follow up on.

Please see our response to the comment above.

2) I appreciate the authors repeated the proteomic analysis in PMA-differentiated THP-1 macrophage-like cells. The manuscript will greatly benefit from a more comprehensive analysis comparing the data generated in HFFF-TERTs and THP-1 to elucidate potential common and cell type-specific virus-regulated proteins.

Many thanks for this point. We have already provided a comprehensive analysis comparing data generated in HFFF-TERTs and THP-1s. Presently, this is in the form of: (a) detailed comparison of proteins that change commonly in both cell types and cell-type specific changes (**Figures 3d, 3f**); (b) enrichment analysis of the types of protein that change in both scenarios (**Figure 3d, Supplementary tables 2-3**); (c) detailed analysis of both of the above in the main text and discussion. However, for maximal value to the community, we agree that it would be useful to highlight the comparison between these cell types, and have generated a new section of results “Comparison of protein changes during MVA infection in fibroblasts and macrophages”. In this, we now incorporate a new table identifying the specific proteins that change commonly or in a cell-type specific manner (**Supplementary Table 7**).

3) While they analyzed similarities and difference carefully between the cell types as well as previous similar studies, it seems to end in a list of their observation. The discussion lacks significance of factors they found from the perspective of vaccine design and development.

Many thanks for this suggestion. We have now addressed this point comprehensively as discussed above. In the discussion, we specifically refer to the significance of our findings with regards to vaccine design in two ways. First, we discuss the finding about ISG20 restriction of viral gene expression in the context of the well-characterised capacity of MVA vectors to attain high levels of heterologous gene expression under the control viral promoters. Second, we discuss the findings related to pyroptosis in the context of vaccine immunogenicity, since inflammasome-dependent release of cytokines and antigens derived from pyroptotic cells shape adaptive antigen-specific immunity.

4) Systematic errors and incorrect peptide assignment increasing the false peptide identification can occur in high-throughput analysis due MS/MS fragmentation patterns. The authors suggested to maximize coverage, the dataset was not filtered to eliminate proteins with single peptide quantification. Since this manuscript is the first global view of the impacts of MVA infection on the host proteome, and low number of biological replicates, to avoid false positive rates, more conservative criteria should be applied ≥ 2 unique peptides to be identified within a single protein for its positive identification.

Many thanks for this point. We agree with the reviewer that it is important to consider the confidence in protein identifications. Our Supplementary Table 1 already includes the number of independent peptide sequencing events for each protein (across the two replicates, 1– 2088 peptides/protein, with 89% of proteins quantified by 2 or more peptides). We have also included an additional supplemental table 5 showing the sequences, SEQUEST XCorr scores and signal:noise values for all peptides that were quantified. Of note, since each protein identification is generally a result of multiple peptide identifications, it is correspondingly less likely that any given ID will be incorrect. Furthermore, as we employ additional quantitation-level filtering, our false discovery rate for protein identification was actually 0.57%, which is less than the prevailing standard in the field of 1%. This was calculated from the number of quantified reverse hits / total number of proteins quantified. We already provide details of filtering criteria according to XCorr score and signal:noise ratio in the Methods section.

Whilst we agree that the most confidently identified proteins are identified by ≥ 2 peptides, there is nevertheless value in the quantification of proteins identified by fewer peptides, both as potential 'hits' where applicable, to support changes identified in enrichment analysis, and to provide as comprehensive as possible a resource to the community. Furthermore, another disadvantage of completely removing single peptide identifications is the consequent bias against including smaller proteins (since they produce fewer peptides eligible for detection). As such, we have now highlighted proteins quantified by single peptides only in Table S1. Additionally, in our 'plotter', when a protein is only quantified by a single peptide, we highlight this to warn readers that there is a small increase in the potential for an incorrect identification or suboptimal quantitation.

Minor points:

1) TMT reagents have different isotope impurities that need to be include for database search to correct the reporter ion ratio interference across different TMT channels.

Many thanks for this point – we have now included these values in Supplementary Table 6.

Reviewer #6 (Remarks to the Author):

In this manuscript, the authors conducted a multiplexed proteomic analysis of MVA and host at five time points throughout MVA infection of human cells. The experiment included inactivated controls to reveal the contribution of the viral particle with no-, or limited viral gene expression, and might provide a global view of the impact of MVA infection on the host proteome.

As for the major points mentioned by reviewer #4 that the low number of biological replicates were used, a further proteomic analysis in differentiated THP-1 cells was performed in revision, and differences in regulation between cell types were discussed. As for the minor points mentioned by reviewer #4, appropriate modifications have been made in this revision. Furthermore, the meaning of the dots color should be presented clearly in the legends of Figure 2d and 4d.

Many thanks for this point – we have now addressed the meaning of the dot colour in the relevant legends.

Reviewers' Comments:

Reviewer #2:

Remarks to the Author:

I appreciate the author's attempt to address the concerns from the last revision. While some of these new analyses are compelling, including comparison between THP-1 and HFFF-TERTs, the study still lacks significant mechanistic insights, including how deregulated proteins impact replication and response to vaccine/protection. The validation studies cited by the authors remain superficial, and do not provide any concrete insights on the importance of these proteins either in the context of host-pathogen interactions or immune responses to vaccines.

Specific comments:

- 1) The small gain of function screen only finds 1 protein that might be important for replication (ISG20). No further insight into how it is regulated (besides a plausible hypothesis), how it restricts replication (both life cycle staging and mechanism of inhibitor), or how this regulation impacts vaccine induced protection is provided. Loss of function studies should also be conducted to confirm these findings
- 2) No further data is provided on the NUPs, which did not have a readout in the GOF screens. There is no validation that these factors (or any besides ISG20) are important for replication or vaccine mediated protection
- 3) The induction of the inflammasome is a relatively trivial finding. Molecular insights into how the virus induces this pathway, and its contribution to vaccine induced protection would be important to understand the role of this pathway in immune protection.

REVIEWERS' COMMENTS

Reviewer #2 (Remarks to the Author):

I appreciate the author's attempt to address the concerns from the last revision. While some of these new analyses are compelling, including comparison between THP-1 and HFFF-TERTs, the study still lacks significant mechanistic insights, including how deregulated proteins impact replication and response to vaccine/protection. The validation studies cited by the authors remain superficial, and do not provide any concrete insights on the importance of these proteins either in the context of host-pathogen interactions or immune responses to vaccines.

Specific comments:

1) The small gain of function screen only finds 1 protein that might be important for replication (ISG20). No further insight into how it is regulated (besides a plausible hypothesis), how it restricts replication (both life cycle staging and mechanism of inhibitor), or how this regulation impacts vaccine induced protection is provided. Loss of function studies should also be conducted to confirm these findings

Reply: We have added further detail to acknowledge the limitations of our findings in the results section:

“Although ectopic expression of ISG20 limited virus-driven GFP expression, loss-of-function studies will need to be conducted to confirm this phenotype.”

2) No further data is provided on the NUPs, which did not have a readout in the GOF screens. There is no validation that these factors (or any besides ISG20) are important for replication or vaccine mediated protection

Reply: We dedicate an entire paragraph in the discussion for the implications of NUP downregulation for MVA infection. The detailed investigation of the implications of NUP downregulation for vaccine-mediated protection is beyond the scope of this study because it would require *in vivo* experiments. Regarding the importance of NUPs for MVA replication, testing this in human cells is challenging in human cells that do not support the productive replication of MVA. It is unlikely the gain-of-function or loss-of-function of single host factors account for MVA restriction in human cells. The paragraph states:

“A total of 101 proteins were downregulated during MVA, but not VACV-WR infection. These included multiple components of the nuclear pore complex, whose downregulation is likely to disrupt nucleocytoplasmic transport⁶⁰. The innate immune response to virus infection requires an intact NPC to mediate nucleocytoplasmic transport of transcription factors and mRNAs and therefore, downregulation/degradation of NPC proteins may be a viral immune evasion strategy. Although some viral proteases inactivate nucleoporins (NUPs) by cleavage, NUP abundance did not vary substantially during infection by multiple human pathogenic viruses, such as human cytomegalovirus, herpes simplex virus type, influenza A virus, Epstein-Barr virus, SARS-CoV-2 and HIV. Three of the NUPs down-regulated by MVA (NUP54, NUP62, NUP88) were identified as host factors necessary for VACV-WR morphogenesis, raising the possibility that NUP downregulation contributes to arrested virion maturation during MVA infection of human cells. It remains to be determined whether this MVA-induced modulation is shared with the parental strain (CVA) or was acquired during the serial passage in chicken embryo fibroblasts. The molecular mechanisms underpinning restriction of MVA in human cells are not fully understood but are largely attributed to the loss or disruption of genes encoding host-range factors. Nucleoporins have been implicated in antiviral defences, including as co-factors in the MX2-mediated restriction of HIV-1 or as substrates for proteolytic cleavage by the antiviral restriction

factor FAM111A, which restricts SV40 by disrupting the nuclear pore complex. In line with our observations, a recent report confirmed that FAM111A cleaves NUP62 and disrupts nuclear barrier during MVA infection, in addition to inducing the degradation of the viral single-stranded DNA-binding (SSB) protein I3. The antiviral functions of FAM111A are antagonized by the poxvirus host-range factor SPI-1, which is missing in MVA. Restoration of SPI-1 in MVA rescues virus replication in human cells. In our gain-of-function screen, ectopic expression of NUP54, NUP62 or NUP88 did not significantly affect MVA gene expression, possibly because the rescue of individual nucleoporins is insufficient to restore the functional consequence of the simultaneous downregulation of several nuclear pore complex components.”

Because this already provides a significant amount of detail, we have not extended this discussion further.

3) The induction of the inflammasome is a relatively trivial finding. Molecular insights into how the virus induces this pathway, and its contribution to vaccine induced protect would be important to understand the role of this pathway in immune protection.

Reply: We disagree that these findings are relatively trivial. Indeed, presentation of our data at the “Infectious Diseases Through an Evolutionary Lens” conference held just earlier this week in London provoked a significant amount of interest from multiple conference participants, and we have detailed why this is of particular interest in the text already. To fully test the implications of inflammasome activation on vaccine efficacy would require *in vivo* experiments that are beyond the scope of our study. However, we have already covered this point in the discussion and highlighted that “it remains to be determined how activation of the inflammasome and pyroptosis contribute to the immunogenicity and protection of MVA-based vaccines *in vivo*.”